# The LPS-inducible lncRNA Mirt2 is a negative regulator of inflammation

Meng Du[1,2], Lin Yuan[1,2], Xin Tan[1,2], Dandan Huang[1,2], Xiaojing Wang[3], Zhe Zheng[1,2], Xiaoxiang Mao[1,2], Xiangrao Li[1,2], Liu Yang[1,2], Kun Huang[1,2], Fengxiao Zhang[1,2], Yan Wang[1,2], Xi Luo[1,2], Dan Huang[1,2] & Kai Huang[1,2]

Toll-like receptors (TLRs) are a family of pattern recognition receptors (PRR) with a crucial function in innate immune responses. Activation of TLR4 signaling at the plasma membrane by lipopolysaccharide (LPS) stimulates proinflammatory signaling pathways dependent on the E3 ubiquitin ligase TRAF6. Here we show the LPS-induced long non-coding RNA (lncRNA) Mirt2 functions as a checkpoint to prevent aberrant activation of inflammation, and is a potential regulator of macrophage polarization. Mirt2 associates with, and attenuates Lys63 (K63)-linked ubiquitination of, TRAF6, thus inhibiting activation of NF-κB and MAPK pathways and limiting production of proinflammatory cytokines. Adenovirus mediated gene transfer of Mirt2 protects mice from endotoxemia induced fatality and multi-organ dysfunction. These findings identify lncRNA Mirt2 as a negative feedback regulator of excessive inflammation.

[1] Department of Cardiology, Union Hospital, Tongji Medical College, Huazhong University of Science and Technology, Wuhan 430000, China. [2] Clinic Center of Human Gene Research, Union Hospital, Tongji Medical College, Huazhong University of Science and Technology, Wuhan 430000, China. [3] Department of Pediatric Surgery, Union Hospital, Tongji Medical College, Huazhong University of Science and Technology, Wuhan 430000, China. Correspondence and requests for materials should be addressed to K.H. (email: unionhuang@163.com)

Innate immune responses have the capacity to both combat infectious microbes and drive pathological inflammation, which contributes to diseases such as sepsis, atherosclerosis, obesity, autoimmunity and cancer[1–3]. Toll-like receptors (TLR) are pattern recognition receptors (PRR) in the innate immune system, and each TLR recognizes specific pathogen-associated molecular patterns (PAMP)[4]. Lipopolysaccharide (LPS) is a natural adjuvant synthesized by Gram-negative bacteria that stimulates cells through TLR4, and has profound effects on immune responses[5]. TLR4-triggered signaling depends on the adaptor proteins myeloid differentiation marker 88 (MyD88) and Toll–interleukin-1 (IL-1) receptor (TIR) domain–containing adaptor-inducing IFNβ (TRIF), which mediate distinct responses that are classified as MyD88-dependent and TRIF-dependent signaling pathways[6].

At the plasma membrane, the binding of MyD88 to TLR4 results in the recruitment and phosphorylation of IL-1 receptor-associated kinase 1 (IRAK1) and IRAK4, which facilitate oligomerization and auto-ubiquitination of TNF receptor–associated factor 6 (TRAF6)[7, 8]. Ubiquitinated TRAF6 subsequently engages other signaling proteins, such as transforming growth factor β–activated kinase (TAK1), to activate the inhibitor of κB (IκB) kinase (IKK) and mitogen-activated protein kinase (MAPK) kinase (MKK), leading ultimately to activation of transcription factors such as nuclear factor kappa B (NF-κB) and activator protein 1 (AP-1) to induce immune and inflammatory responses[9, 10].

Long non-coding RNAs (lncRNA) are a large class of non-protein-coding transcripts that are greater than 200 bases in length[11]. They are involved in many physiological and pathological processes that include genomic imprinting, embryonic development, cell differentiation, tumor metastasis and regulation of the cell cycle[12–14]. Although a number of lncRNAs have been reported to have crucial functions in diverse processes and diseases, only a few lncRNAs have been show to regulate the immune system[15–17].

In this study, we investigate global lncRNA expression profiles using microarray analysis of macrophages treated with LPS, and propose a model whereby TLR signaling induces the up-regulation of lncRNA-Mirt2, which serves as a repressor of inflammatory responses through interaction with TRAF6, and inhibition of its oligomerization and auto-ubiquitination.

## Results

**Differentially expressed lncRNAs in LPS-activated macrophages**. To identify the lncRNAs that are involved in the innate immune response, we performed a microarray analysis in primary cultured peritoneal macrophages obtained from C57BL/6 mice. LPS, which is a TLR4 ligand, induced numerous differentially expressed lncRNAs. In the volcano plot, 64221 lncRNAs were represented, of which, 2070 were significantly upregulated (red plots) and 1750 were downregulated (blue plots) when filtered with a threshold of a fold change $\geq 2$ and $q < 0.05$ (Fig. 1a). Differentially expressed lncRNAs, between the control and LPS treatment group, were explored by using more stringent criteria (Student's $t$ test, $P < 0.01$, fold change $> 20$) and filtered according to transcript abundance. In this way we identified the 145 lncRNAs that were most highly induced by LPS stimulation, of which 98 were upregulated and 47 were down regulated (Fig. 1b). LncRNA-Mirt2 was among the most highly induced upregulated lncRNAs and was abundantly expressed in macrophages.

**Macrophage Mirt2 is induced by LPS and repressed by IL-4**. The response of lncRNA-Mirt2 to TLR4 signaling was confirmed by qRT-PCR. Mirt2 expression in cultured peritoneal macrophages was induced by LPS in a time- and dose-dependent manner, which peaked at 10 h at a concentration of 1 μg/mL (Fig. 1c, d). The cell viability was confirmed using the MTT assay. Fluorescence in situ hybridization (FISH) showed that Mirt2 was primarily located in the cytoplasm (Fig. 1e), suggesting that Mirt2 might exert its biological function in the cytoplasm. Surprisingly, the increase in Mirt2 was not macrophage- or TLR4 signaling specific. As demonstrated in Supplementary Fig. 1a, LPS stimulation also induced obvious Mirt2 upregulation in tracheal epithelial cells, hepatocytes and smooth muscle cells. In addition to responding to macrophage TLR4 signaling through LPS stimulation, Mirt2 was also induced by Pam₂CSK₄ (a TLR2/6 agonist) and R848 (a TLR7/8 agonist) as well. Conversely, Pam₃CSK₄ (a TLR1/2 agonist) and Poly (I:C), which is a synthetic double-stranded RNA (a TLR3 agonist), had no effects on Mirt2 expression (Supplementary Fig. 1b). The immune phenotype of macrophages depends on the cellular environment and the presence of various activator molecules. Proinflammatory molecules, such as interferon-γ (IFN-γ) and LPS, result in classical activation of macrophages (M1 macrophage). In contrast, alternatively activated macrophages (M2 macrophage), which differentiate upon IL-4 stimulation, exhibit a different phenotype that provokes tolerance or T helper 2 (Th2) immune responses. Notably, IL-4 stimulation resulted in a dramatic decrease in Mirt2 expression that was significant at 2 h and most obvious at 6 h (Supplementary Fig. 1c). These findings led to the hypothesis that Mirt2 was a regulator of macrophage polarization. Analysis of the tissue distribution in mice revealed that Mirt2 was abundantly expressed in macrophages (peritoneal and alveolar), followed by the kidney, lung, and liver (Fig. 2a). Consistent with our in vitro experiment results, LPS administration induced a time-dependent increase in the Mirt2 levels in various mouse tissues, including the heart, liver, spleen, lung, kidney, fat, vessels, peritoneal macrophages, alveolar macrophages and serum (Fig. 2b). Similar to the colocalization analysis, which demonstrated colocalization in the adipose tissue and lung, a marked increase in the Mirt2 signal was detected within the CD68-positive macrophages from the LPS-treated mice (Fig. 2c). These data show that the Mirt2 level can vary dynamically according to the macrophage polarization state, the two extremes of which are induced by LPS and IL-4.

**Upstream mediators of Mirt2 expression in macrophages**. LPS acts through distinct pathways leading to macrophage activation. Selective pharmacological inhibitors (Bay-11-7082 (NF-κB), U0126 (extracellular signal-regulated kinase, Erk), SP600125 (c-Jun N-terminal kinase, Jnk), SB203580 (p38), and PF-04965842 (Janus kinase 1, Jak1)) were used to assess the functional consequences of inhibition of these pathways on Mirt2 expression. Pretreatment with SB203580 completely abrogated LPS-induced Mirt2 upregulation in macrophages. In addition, the selective Jak1 inhibitor PF-04965842 exerted partial inhibitory effects on LPS-induced Mirt2 expression (Fig. 3a).

To further explore the mechanisms underlying the Mirt2 elevation in LPS-activated macrophages, the 4000 bp Mirt2 promoter-luciferase reporter construct and its truncations were established and transfected into cultured RAW264.7 cells. LPS treatment significantly increased the luciferase activity of all truncated constructs, including the shortest construct, which was expected to include the potential LPS response element (Fig. 3b). We further performed in silico analysis of this sequence (−389 bp to +163 bp) and found two separated signal transducer and activator of transcription 1 (Stat1) binding sites. Silencing of Stat1 completely abrogated LPS-induced Mirt2 expression in macrophages, whereas an opposite effect was found for Stat1 over-

expression (Fig. 3c). The knockdown efficiency for the Stat1-siRNA was shown by Western blot (Fig. 3d). In line with these results, mutation of either of the two sites partially reduced the luciferase activity induced by LPS, and mutation of both sites completely reversed these effects (Fig. 3e). These results demonstrated that the −389 bp to +163 bp elements were

involved in LPS-dependent Mirt2 upregulation and that this response was mediated by the transcription factor Stat1. Our previous studies demonstrated that inhibition of the p38 pathway by SB203580 induced a reduction in Mirt2 expression in response to LPS. Consistently, enforcing p38 expression increased the Mirt2 levels in macrophages either under resting conditions or

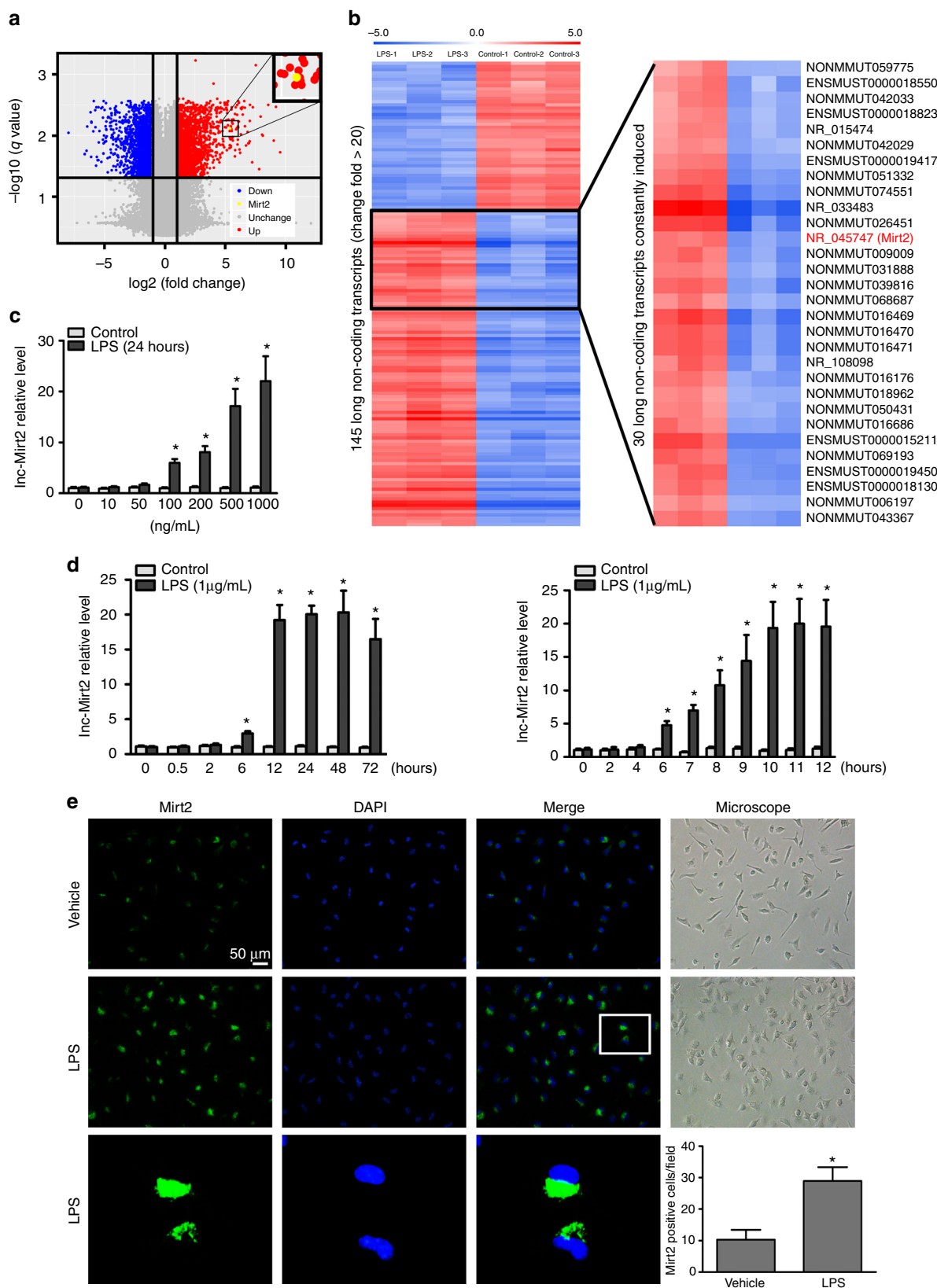

after activation by LPS, whereas these effects could be completely abrogated by knockdown of Stat1 (Fig. 3f). Inducing p65 ERK1, ERK2, JNK1 and JNK2 expression had no effect on Mirt2 expression (Supplementary Fig. 1d). These results indicated that the LPS-p38-Stat1 pathway was responsible for the LPS-induced upregulation of Mirt2 in macrophages. In addition to the p38 pathway, LPS-induced autocrine/paracrine IFN-α/β also mediates the activation of Stat1, which cannot be inhibited by the p38 inhibitor SB203580. As demonstrated in our study, IFN-β stimulation did not affect the basal Mirt2 level but further increased its expression in LPS-activated macrophages, and these effects could be completely abolished by Stat1 silencing (Fig. 3g). Previous studies have shown that p38 is involved in the induction of IFN-β in macrophages[18, 19,]. However, supplementation with exogenous IFN-β did not rescue the inhibitory effects of the p38 inhibitor on Mirt2 expression (Fig. 3h). Thus, the decrease in Mirt2 expression is not due to decreased IFN-β after p38 inhibitor treatment. Moreover, a neutralizing antibody targeting IFN-α/β partly inhibited the LPS-induced Mirt2 upregulation (Fig. 3i). The results indicated that LPS-p38-Stat1 was indispensable for Mirt2 expression, whereas LPS-IFN-α/β-Stat1 enforced these effects. Therefore, replenishment with exogenous IFN-β did not rescue the inhibitory effects of the p38 inhibitor on Mirt2 expression, since the LPS-p38-Stat1 pathway reminded repressed.

To further explore the mechanism, we evaluated the induction of IFN-β and the activation of Stat1 upon TLR stimulation. Our studies showed that engagement of TLR2 (Pam$_3$CSK$_4$ and Pam$_2$CSK$_4$), TLR4 (LPS) and TLR7/8 (R848) led to rapid phosphorylation (30 min) of Stat1 at serine 727 (S-727) in murine macrophages, whereas TLR4 (LPS), TLR3 (Poly (I:C)) and TLR7/8 (R848) induced Stat1 phosphorylation at tyrosine 701 (T-701), although this response was delayed (4 h) compared with S-727 phosphorylation (Fig. 3k). Previous studies showed that TLR-induced Stat1 serine phosphorylation was dependent on p38; however, tyrosine phosphorylation of Stat1 was indirectly mediated by the production of endogenous type I IFNs, particularly IFN-β[20]. Consistently, we found that TLR4, TLR3 or TLR7/8 stimulation led to robust induction of IFN-β (Fig. 3j). Our previous results showed that only TLR2, TLR4 and TLR7/8 stimulation led to Mirt2 induction (Supplementary Fig. 1b), which indicated that p38-Stat1 (S-727) was indispensable for the induction, whereas IFNα/β-Stat1 (T-701) enforced these effects. The possible mechanisms are depicted in Supplementary Fig. 6.

Moreover, the induction of Mirt2 by LPS was partially inhibited by knockdown of Myd88 or TRIF, and knockdown of both proteins completely abrogated the effects of LPS (Supplementary Fig. 1e). The knockdown efficiencies of the Myd88 and TRIF siRNAs were evaluated by Western blot (Fig. 1f). TRIF signaling is more prominent in type I IFN expression, whereas p38 activation occurs independently through the Myd88 or TRIF pathway[21, 22]. Knockdown of TRIF abrogated the induction of type 1 IFNs but had no effect on p38 phosphorylation. In addition, knockdown of Myd88 did not completely inhibit p38 activation. We considered that knockdown of both proteins could completely abrogate the induction of Mirt2 by LPS.

As demonstrated previously, IL-4 stimulation resulted in a dramatic decrease in Mirt2 expression (Supplementary Fig. 1c).

Jak3, which is a member of the Jak family kinases, was identified as the primary effector molecule coupled with the IL-4 receptor in myeloid cells[23]. Pretreatment with the Jak3 inhibitor WHI-P154 reversed the effects of IL-4, whereas the PI3K inhibitor wortmannin and Jak2 inhibitor TYRPhostin had no effects (Supplementary Fig. 1g–i). Using a series of luciferase reporter constructs containing deleted 5′-flanking regions of Mirt2, we identified the sequence -2324 bp to -1754 bp as an IL-4-response element (Supplementary Fig. 1j). The binding sites of Stat6 and PPARγ, which are crucial downstream signaling molecules of IL-4-Jak3, were both included in this sequence. However, only Stat6 silencing restored the expression of Mirt2 that was inhibited by IL-4 (Supplementary Fig. 1k). The knockdown efficiencies for the Stat6 and PPARγ siRNAs were shown by Western blot (Supplementary Fig. 1l). Consistently, a mutation of the Stat6 binding site reversed the luciferase activity attenuated by IL-4 stimulation, whereas a mutation of the PPARγ binding site had no effect (Supplementary Fig. 1m). These results illustrated that Stat6, which is a key downstream transcription factor of IL-4, negatively regulated Mirt2 expression in macrophages.

**Mirt2 regulates macrophage polarization.** To define the functional role of Mirt2 in LPS-mediated inflammatory and immune responses in macrophages, Mirt2 expression was enforced or silenced by adenovirus-mediated gene modification. Enforcing or silencing Mirt2 expression did not alter the basal levels of inflammatory factors, such as TNF, IL-1β, IL-6, and IL-12. However, in LPS-activated macrophages, ectopic Mirt2 expression inhibited the aberrant activation of all of these inflammatory factors at 4 and 20 h, and Mirt2 silencing led to a further increase in TNF, IL-1β, IL-6, and IL-12 expression at 20 h; at this latter time point, the levels of these cytokines dropped rapidly in the control group (Fig. 4a–d). In contrast, the LPS-induced expression of type I IFNs and IFN-dependent genes, such as IFN-β and interferon-stimulated gene 15 (isg15), were unaffected by Mirt2 enforcement or silencing (Fig. 4e, f).

Because the NF-κB and MAPK pathways were definitely involved in the inflammatory responses in the LPS-activated macrophages, we examined the correlation between Mirt2 and the phosphorylation of critical signaling proteins of these two pathways. As demonstrated in Fig. 4g, the phosphorylation levels of p65, Jnk, Erk1/2 and p38 in macrophages peaked at 0.5 and 2 h after LPS treatment and dropped rapidly to baseline at 6 and 10 h. Surprisingly, exogenous Mirt2 reduced the phosphorylation levels of p65, Jnk and p38, and silencing Mirt2 led to sustained signaling pathway activation for up to 10 h. Consistently, the immuno-fluorescence assay demonstrated that Mirt2 inhibited LPS-induced p65 nuclear translocation in cultured macrophages (Fig. 4h, i). Collectively, these data suggest that Mirt2 acts as a negative regulator to suppress NF-κB and MAPK signaling pathway activation, thereby inhibiting the upregulation of inflammatory genes in LPS-activated macrophages.

In contrast to LPS, IL-4, which facilitates the differentiation of alternatively activated macrophages, decreased the Mirt2 level, as demonstrated previously (Supplementary Fig. 1c). We explored whether Mirt2 was involved in the expression of the classic M2

**Fig. 1** Differentially expressed lncRNAs in macrophages treated with LPS. **a** Primary cultured peritoneal macrophages were treated with LPS (1 μg/mL) for 24 h and then evaluated to determine their lncRNA profiles using an lncRNA expression microarray. A volcano plot showing the relationship between the $q$ values and the magnitude of the differences in the expression values of the samples in the different groups. **b** The cluster heatmap shows lncRNAs with expression change fold > 20 from microarray data ($P < 0.01$). **c** The expression of Mirt2 in cultured peritoneal macrophages treated with LPS of concentration gradient for 24 h. **d** The expression of mirt2 in cultured peritoneal macrophages treated with LPS (1 μg/mL) for different times. **e** Macrophages, treated with LPS or vehicle, were labeled with a lncRNA-Mirt2 probe (green) using Fluorescent in site hybridization (FISH), and counterstained with 4′,6-diamidino-2-phenylindole (DAPI) (nucleus staining, blue). The number of positive cells in the field was quantified. Data represent the mean ± SEM of three independent experiments. *$P < 0.05$ vs. control group. Two-tailed Student's $t$ test for two groups

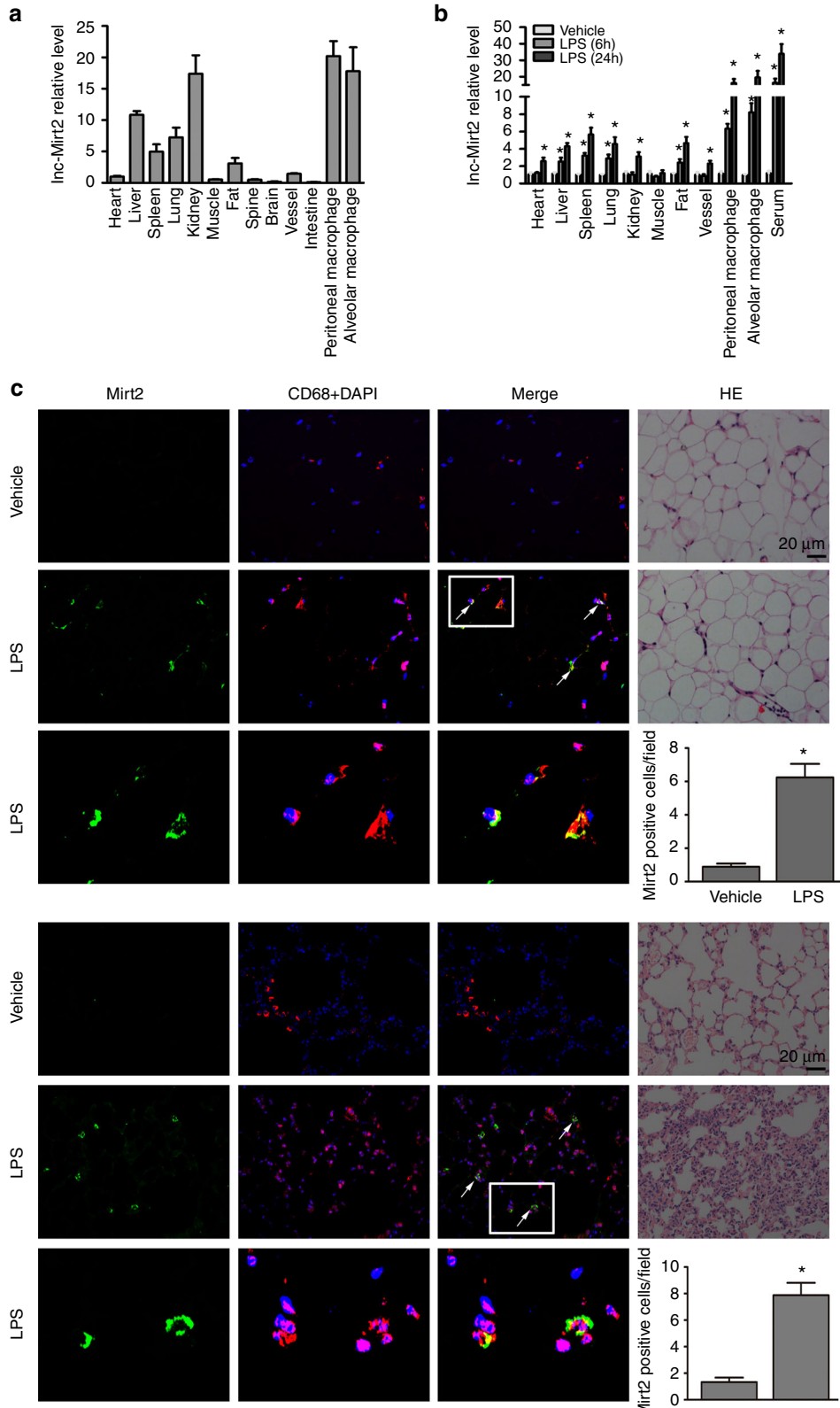

**Fig. 2** Mirt2 expression is induced in mice with endotoxemia. **a** Expression and distribution of Mirt2 in C57BL/6 mice. **b** Expression of Mirt2 in various tissues of endotoxemia mice challenged with LPS (25 mg/kg) for 6 and 24 h. **c** RNA FISH analysis of Mirt2 in adipose tissue (upper) and lung (lower) from normal or mice with endotoxemia. For colocalization analysis, sections were co-stained for Mirt2 (green) and CD68 (red, macrophage marker). DAPI was used for nucleus staining (blue). Arrows indicate Mirt2 and CD68 double positive cells. Double positive cells were counted in four sections per mouse and quantified. Data are expressed as mean ± SEM (n = 6). *P < 0.05 vs. control group Two-tailed Student's *t* test for two groups

markers and cytokines induced by IL-4. In macrophages that ectopically expressed Mirt2, IL-4 stimulation led to a more rapid and remarkable increase in the Arg1, CD206, Fizz1, Ym1, and CCL24 levels, and Mirt2 silencing mirrored the effects of Mirt2 over-expression (Supplementary Fig. 2a–h). However, Stat6 phosphorylation and PPARγ expression, which definitely mediated IL-4-induced macrophage M2 polarization, were unaffected by Mirt2 (Supplementary Fig. 2i). The adenovirus infection efficiency was examined by qRT-PCR (Supplementary

Fig. 2j). These data indicated that in addition to its involvement in regulation of the LPS-induced inflammatory response, Mirt2 also affected the induction of macrophage M2 polarization by IL-4 via a mechanism that was probably independent of the Stat6 and PPARγ pathways.

**Mirt2 inhibits TRAF6 oligomerization and Auto-ubiquitination.** Next, we performed a pull-down assay with biotinylated Mirt2, followed by mass spectrometry (MS) to search

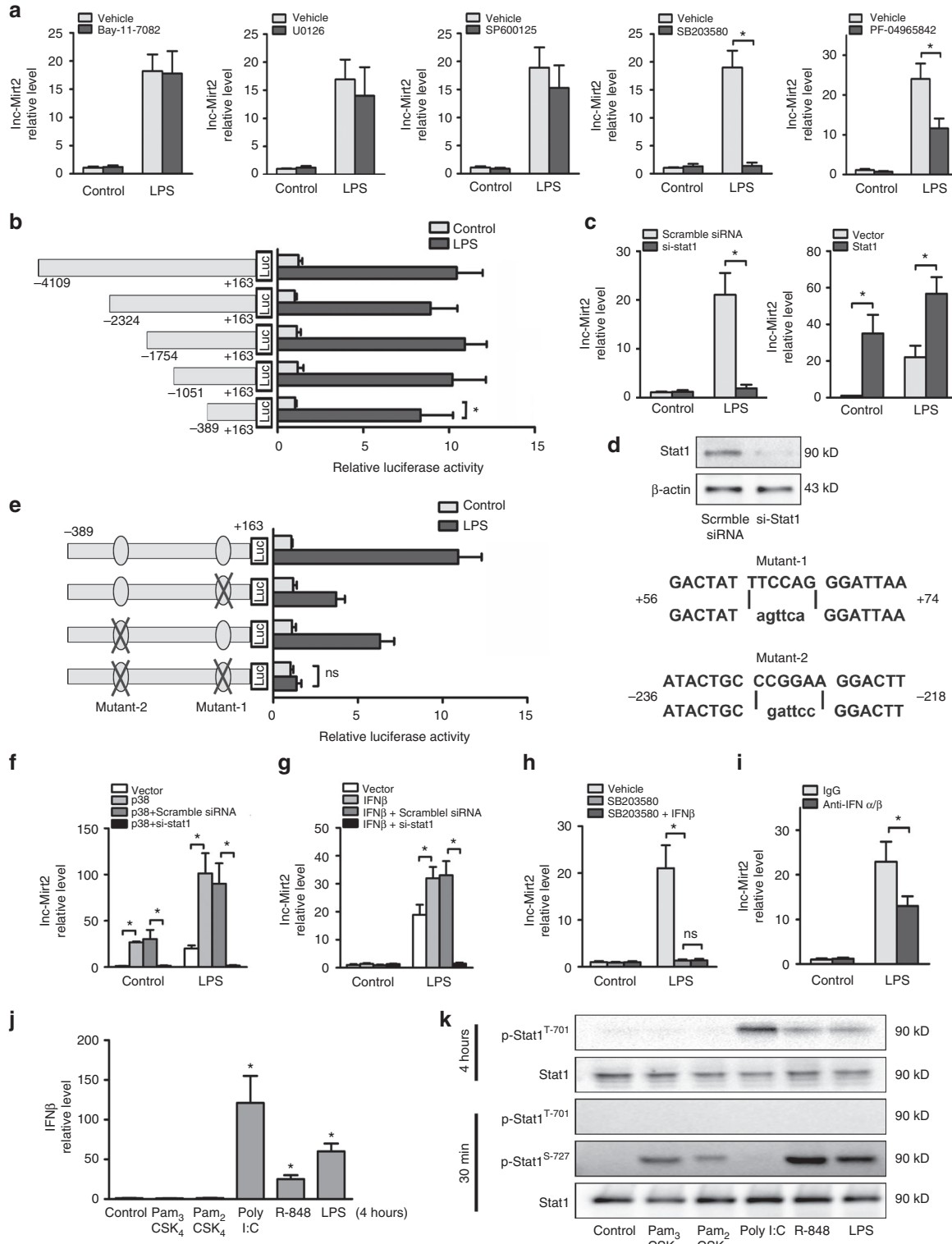

for potential Mirt2-interacting proteins in macrophages. TRAF6, which is a key signaling protein with crucial functions in the activation of the canonical NF-κB and MAPK pathways, was identified as a Mirt2-associated protein (Fig. 5a). This finding was confirmed by independent immunoblot (Fig. 5b), and an RNA immunoprecipitation (RIP) assay further verified the specificity of this interaction (Fig. 5c). Moreover, confocal microscopy for Mirt2 and TRAF6 immunostaining showed that Mirt2 colocalized with TRAF6 in the cytoplasm of cultured macrophages activated by LPS (Fig. 5d). We also observed colocalization between Mirt2 and TRAF6 in non-stimulated macrophages (Fig. 5d), which indicated that the endogenous interaction was not dependent on stimulation. However, since the basal Mirt2 level was relatively low and TRAF6 was activated upon stimulation, the roles of Mirt2 in resting cells could be limited, although this possibility needs further study. RNA pulldown assays with Mirt2 truncations revealed that only full-length Mirt2 bound to TRAF6, whereas the other truncations completely lost their binding capacity (Fig. 5e). TRAF6 consists of an amino terminal RING domain, which is followed by several zinc-finger motifs, a central coiled-coil region and a highly conserved carboxyl terminal domain known as the TRAF-C domain[24]. We performed protein domain mapping experiments using truncated TRAF6, which showed that the RING domain and zinc-finger motifs of TRAF6 were responsible for the interaction with Mirt2 (Fig. 5f).

Upon activation of TLR4 signaling, TRAF6 is recruited to the receptor complexes, wherein it assists TLR4 in mediating downstream signaling, including NF-κB and MAPK. Oligomerization and auto-ubiquitination of TRAF6 mediated by the RING domain and zinc-finger motifs is essential for signal transduction[25]. We determined whether the interaction of Mirt2 with TRAF6 at the TRAF-N domain affected its oligomerization and ubiquitination. Our Co-IP assays showed that the oligomerization of TRAF6 was significantly inhibited by exogenous Mirt2 in a dose-dependent manner (Fig. 5g). To determine the effects of Mirt2 on the ubiquitination of TRAF6, we transfected HEK293T cells to express Myc-tagged TRAF6 and HA-tagged ubiquitin in the presence of Ad-EV or Ad-Mirt2. Then the cells were lysed and subjected to IP with an anti-Myc antibody and IB with an anti-HA antibody. The results showed that the presence of Mirt2 significantly affected the amount of ubiquitin on TRAF6 (Fig. 5h, left panel). In further studies to determine the type of ubiquitin on TRAF6, HEK293T cells were transfected with plasmids expressing mutant forms of ubiquitin that specifically ubiquitinated via either the K48 or K63 linkages. The presence of Mirt2 resulted in less K63-linked ubiquitin in a dose-dependent manner, but no significant change was found in K48-linked ubiquitin (Fig. 5h, middle and right panels). This result indicated that Mirt2 diminished the activating K63-linked ubiquitination of

TRAF6 but left the degradative K48-linked ubiquitination unchanged. This finding was further verified in LPS-treated peritoneal macrophages, in which the K63-ubiquitination of TRAF6 was strongly elevated at 30 min and dropped to baseline levels at 6 h. Mirt2 abrogated the induction of TRAF6 K63-ubiquitination by LPS, and knockdown of Mirt2 led to sustained ubiquitination even at 6 h after LPS treatment (Fig. 5i). In contrast, the auto-ubiquitination of TRAF2 and TRAF3 was unaffected by Mirt2 (Supplementary Fig. 3a–c). TRAF6-mediated Lys63-linked polyubiquitination requires the integrity of the RING domain and its ability to interact with the heterodimeric E2 complex of Ubc13 and Uev1A. Whereas Ubc13 mediates a direct interaction with an E3, Uev1A provides the linkage specificity[26]. As expected, Mirt2 also inhibited the interaction of TRAF6 with Ubc13 in HEK293T cells expressing exogenous TRAF6 and Ubc13 (Fig. 5j) or in macrophages activated by LPS (Fig. 5k). To verify the crucial role of TRAF6 in Mirt2-mediated inflammatory regulation, we knocked down TRAF6 in macrophages and found that the effects of Mirt2 silencing, which led to sustained activation of the NF-κB and MAPK pathways upon LPS stimulation, were completely abolished (Fig. 5l). Additionally, the LPS-TLR4 signaling cascade was affected through the downstream signaling proteins of TRAF6, such as p-TAK1, p-MKK4, p-MKK7, and p-IKKα/β, which were affected by Mirt2 enforcing or silencing, whereas the upstream signaling molecules of TRAF6, such as TLR4, CD14, Myd88, and p-IRAK4, reminded unchanged (Supplementary Fig. 3d–g). Since we failed to detect an interaction between TAK1 and Mirt2 (Supplementary Fig. 3h), we considered that the diminished phosphorylation of TAK1 was due to inhibition of TRAF6 ubiquitination by Mirt2. In addition to the Toll-like receptor family, TRAF6 is also a crucial docking molecule that mediates signaling events initiated by the interleukin-1 (IL-1) and tumor necrosis factor (TNF) receptor families (i.e., receptor activator of NF-κB, RANK) in macrophages[25, 27]. As demonstrated in Supplementary Fig. 3i, the phosphorylation of p65, which is activated by IL-1β, or the receptor activator of NF-κB ligand (RANKL, also known as OPGL or ODF) was significantly inhibited by Mirt2. However, Mirt2 had no effect on the TNF-induced activation of p65, which was mediated by TRAF2. These results identified TRAF6 as the key point for Mirt2-mediated inflammatory regulation. Thus, Mirt2 interacted with the RING domain and zinc-finger motifs of TRAF6 and disrupted its oligomerization and association with Ubc13, thereby inhibiting the auto-ubiquitination of TRAF6 and related inflammatory responses. The diminished ubiquitination of TRAF6 did not seem to be due to the functions of de-ubiquitination enzymes, since A20 expression was inhibited and CYLD reminded unchanged upon Mirt2 over-expression (Supplementary Fig. 3j and k).

**Fig. 3** LPS upregulates Mirt2 in macrophages via p38-Stat1 and IFN-Stat1 signaling. **a** Cultured peritoneal macrophages were pretreated with selective pharmacological inhibitors Bay-11-7082 (NF-κB, 10 μM), U0126 (Erk, 50 μM), SP600125 (Jnk, 50 μM), SB203580 (p38, 50 μM) or PF-04965842 (Jak1, 50 nM), then cells were activated by LPS (1 μg/mL, 24 h). Expression of Mirt2 was detected by qRT-PCR. **b** Luciferase reporter constructs containing Mirt2 promoter or its truncations were co-transfected with an internal control plasmid pRL-TK into RAW264.7 cells, followed by LPS challenge (1 μg/mL, 24 h). The relative luciferase activities are expressed as a percent of values determined in control group. **c** Effects of Stat1 overexpression or Stat1 knockdown on the expression of Mirt2 in cultured peritoneal macrophages. **d** The knockdown efficiency for Stat1 siRNA. **e** Luciferase reporter constructs containing Mirt2 promoter (-389bp ~ +163 bp) or its mutants (for Stat1 binding sites) were co-transfected with an internal control plasmid pRL-TK into RAW264.7 cells, followed by LPS challenge (1 μg/mL, 24 h). The left panel demonstrates the relative luciferase activities, which are expressed as a percent of values determined in control group. The right panel demonstrates the mutation for two separate Stat1 binding sites. **f** Effects of p38 overexpression on the expression of Mirt2 in cultured peritoneal macrophages pretreated with scramble siRNA or si-Stat1. **g** Effects of IFN-β on the expression of Mirt2 in cultured peritoneal macrophages pretreated with scramble siRNA or si-Stat1. **h** Effects of exogenous IFN-β on Mirt2 expression inhibited by p38 inhibitor. **i** Cultured peritoneal macrophages were pretreated with antibodies against IFN-α/β, then cells were activated by LPS (1 μg/mL, 24 h). Expression of Mirt2 was detected by qRT-PCR. **j** The expression of IFN-β in cultured peritoneal macrophages treated with TLR ligands. **k** The phosphorylation of Stat1 induced by TLR ligands. Data represent the mean ± SEM of three independent experiments. *$P < 0.05$. ns none significant. Two-tailed Student's t-test for two groups and one-way ANOVA for multiple groups

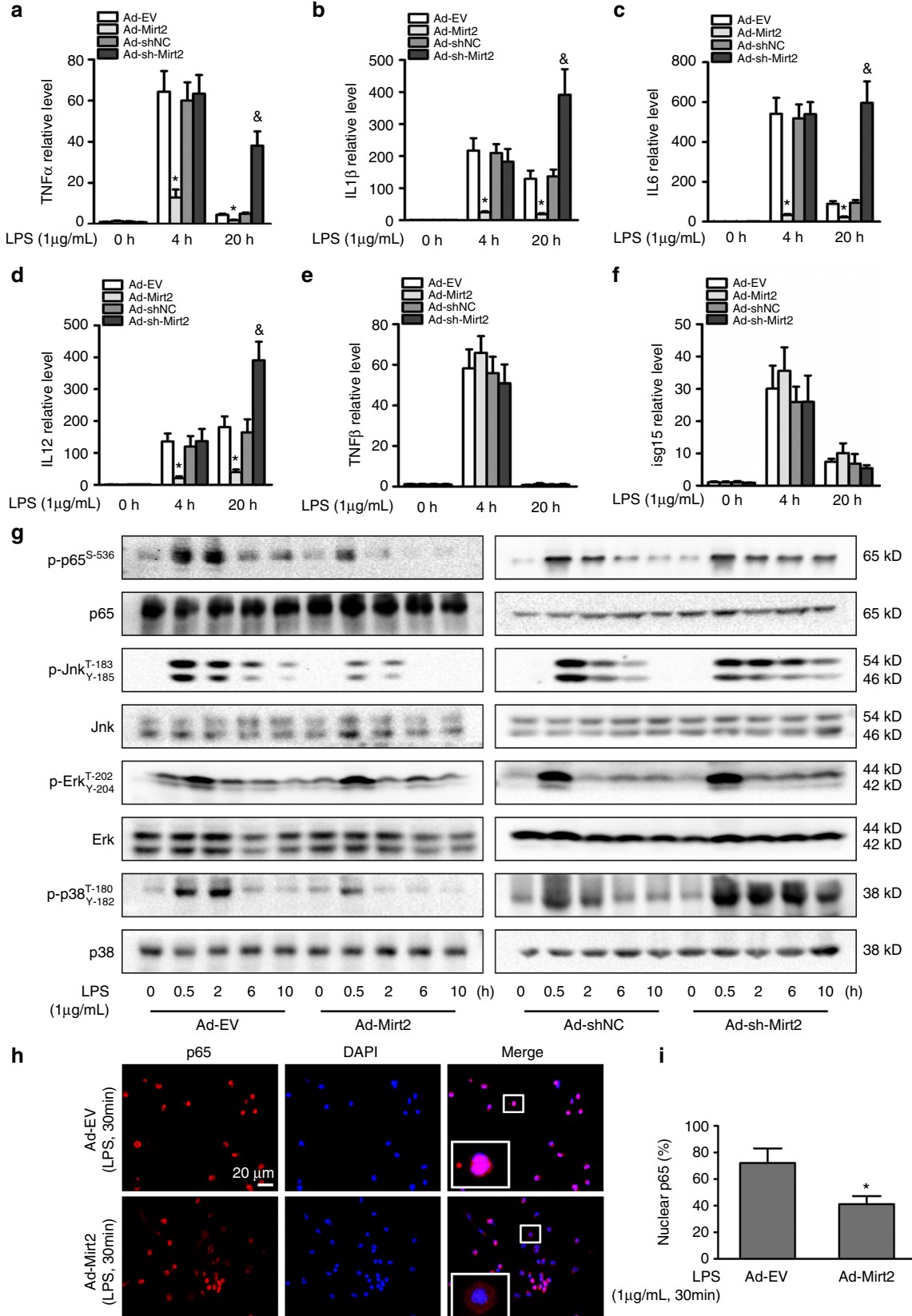

**Fig. 4** Mirt2 inhibits LPS-induced inflammatory responses in macrophages. **a–f** Effects of Mirt2 overexpression or Mirt2 knockdown on the expression of inflammatory factors induced by LPS in cultured peritoneal macrophages, as determined by qRT–PCR. Data represent the mean ± SEM of three independent experiments. *P < 0.05 vs. Ad-EV or Ad-shNC group. **g** Effects of Mirt2 overexpression or Mirt2 knockdown on the activation of NF-κB and MAPK pathways induced by LPS in cultured peritoneal macrophages, as determined by Western blot. **h** Effects of Mirt2 on p65 nuclear translocation in LPS-treated macrophages, assayed by immunofluorescent detection. **i** Quantification of p65 nuclear translocation in panel **h**. Data represent the mean ± SEM of three independent experiments. *P < 0.05 vs. Ad-EV. Two-tailed Student's t-test for two groups and one-way ANOVA for multiple groups

**Mirt2 protects against LPS-induced endotoxemia.** Sepsis, septic shock, and endotoxemia are systemic inflammatory states that affect most organs of the body, including the lung, liver, heart, and white adipose tissue[28]. We investigated the effects of Mirt2 in mice with endotoxemia induced by LPS challenge. Mirt2

recombinant adenovirus administration via the tail vein resulted in a dramatic increase in the Mirt2 levels in various tissues in the mice, especially the liver, followed by the fat, lung, heart, and peritoneal macrophages (Fig. 6a). At 72 h post-LPS injection, only 12.5% of the Ad-EV treated mice survived, whereas Ad-Mirt2

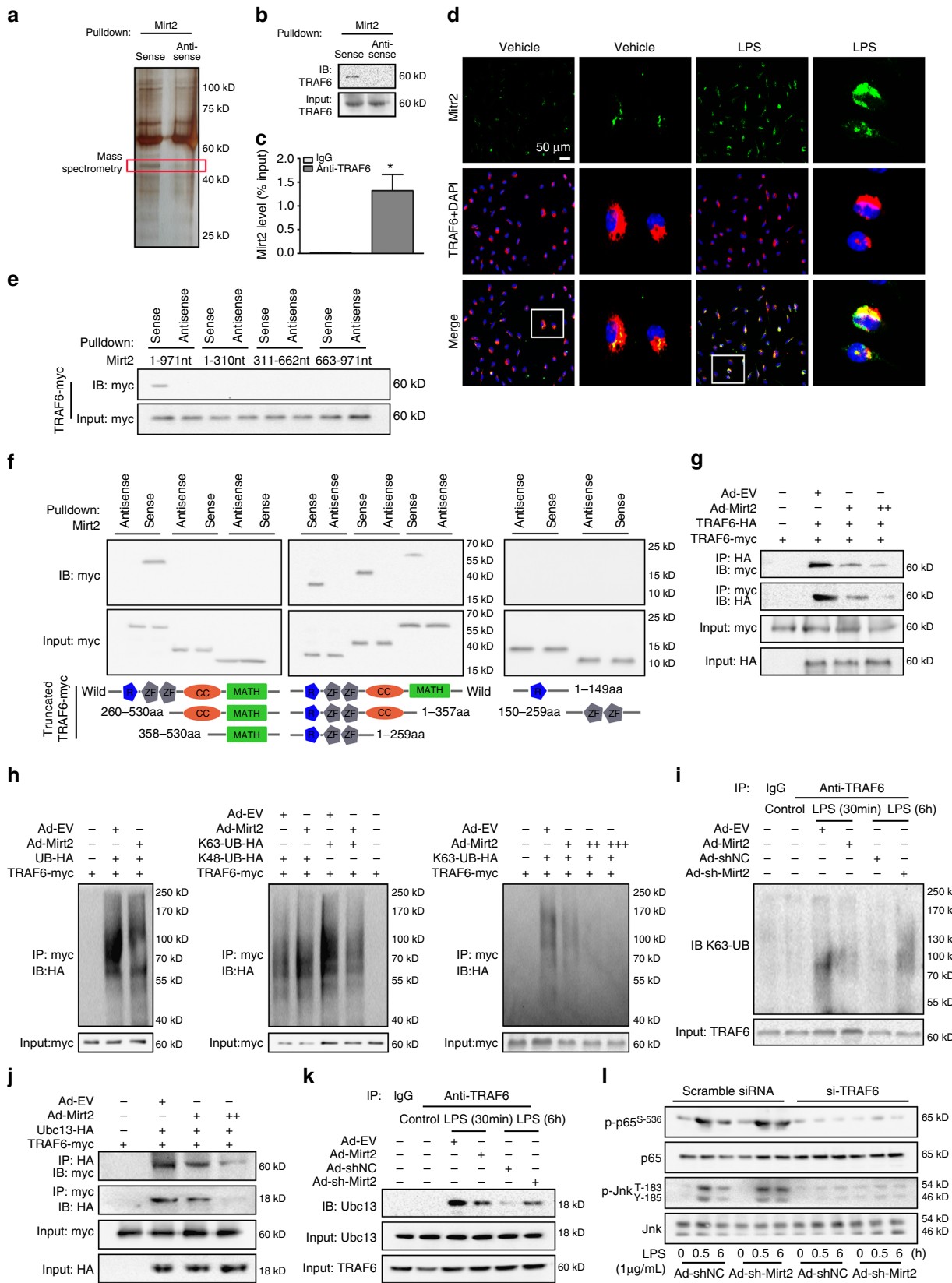

administration showed remarkable protection and led to 50% survival at 72 h in all mice (Fig. 6b). Moreover, exogenous Mirt2 resulted in a significant reduction in the plasma concentrations of the pro-inflammatory cytokines IL-1β, IL-6, and TNF at 6 h post-LPS administration (Fig. 6c). Endotoxemia is associated with organ injury and function failure. Mice challenged with LPS had various degrees of acute inflammation in the lung, liver, heart, and adipose tissues. An examination of the pathology of the lung tissue showed that the LPS-treated mice developed exacerbated lung inflammation, hemorrhaging, and alveolar septal thickening, whereas fewer lung lesions were observed in the mice receiving Ad-Mirt2, with a lower lung injury score and decreased inflammatory cell infiltration (Fig. 6d, e). Western blot assays showed that Ad-Mirt2 pretreatment significantly suppressed LPS-induced p65 and Jnk phosphorylation in the lung tissues compared with the phosphorylation levels in the Ad-EV treated group (Fig. 6f, g).

Consistently, the LPS-induced liver injury and activation of inflammatory signaling pathways were also alleviated in the Ad-Mirt2 treated mice. In contrast, the lipid contents were unaffected, as demonstrated by oil red O staining and determination of the liver triglyceride levels (Fig. 6h–k). The effects of Mirt2 on inflammatory responses in the adipose tissues and hearts of mice with endotoxemia were not as significant as the effects in the lung and liver. Exogenous Mirt2 did not suppress macrophage infiltration and p65 pathway activation in the adipose tissue and heart; however, weakened Jnk phosphorylation was observed in the adipose tissue in the mice treated with Ad-Mirt2 (Supplementary Fig. 4a–h). Together, these results showed that Mirt2 could effectively relieved LPS-induced endotoxemia and organ dysfunction, especially in the lung and liver. Furthermore, the anti-inflammatory effects of Mirt2 were confirmed in ex vivo cultured macrophages from mice receiving adenovirus. The LPS-induced expression of inflammatory factors in macrophages from the Ad-Mirt2 treated mice was significantly inhibited compared to expression in macrophages from the control group (Supplementary Fig. 4i–k).

**Mirt2 is anti-inflammatory in various cell types.** We previously demonstrated that Mirt2 alleviated inflammatory responses in macrophages by inhibiting TRAF6 oligomerization and auto-ubiquitination. However, whether these effects are macrophage-specific or universal in other cell types remains incompletely understood. To clarify this point, primary cells (tracheal epithelial cells, hepatocytes, adipocytes, cardiomyocytes, and cardiac fibroblasts) representing four important organs damaged during endotoxemia were cultured and challenged with LPS. Ad-Mirt2 pretreatment significantly inhibited LPS-induced IL-6, IL-1β, and

IL-23α expression in tracheal epithelial cells (Fig. 7a). Consistent with our observations in macrophages, p65 and Jnk phosphorylation and the K63-ubiquitination of TRAF6 were abrogated by exogenous Mirt2 (Fig. 7b–d). Similarly, in LPS-treated primary hepatocytes, the aberrant activation of the inflammatory factors IL-6, MCP1 and CXCL9 was restrained by Mirt2 over-expression (Fig. 7e). However, Mirt2 had no apparent effects on the expression of inflammatory factors and activation of the NF-κB and MAPK pathways induced by LPS in adipocytes (Supplementary Fig. 5a–c), cardiomyocytes (Supplementary Fig. 5d) or cardiac fibroblasts (Supplementary Fig. 5e). These results showed that the effects of Mirt2 on inflammatory regulation were not macrophage-specific and were also found in at least tracheal epithelial cells and hepatocytes. This finding may partially explain why Ad-Mirt2 palys a more prominent protective role in the lung and liver in mice with endotoxemia. Surprisingly, an adenovirus carrying the murine origin mirt2 gene also exerted significant anti-inflammatory effects in some cell types derived from humans, such as monocyte-derived macrophages (Supplementary Fig. 5f) and hepatocytes (Supplementary Fig. 5g).

**Involvement of Mirt2 in inflammatory diseases.** Given the important roles of Mirt2 in regulating the inflammatory processes in mice with endotoxemia, we were interested in investigating whether Mirt2 was involved in other inflammatory diseases. Surprisingly, the Mirt2 levels were elevated in the aortas of atherosclerotic mice, the myocardia of AMI (acute myocardial infarction) mice, the adipose tissues of ob/ob mice, and the kidneys and myocardium of STZ (streptozotocin)-induced diabetic mice compared with the levels in their control littermates (Fig. 8a, b, f–h). In contrast, for mice with CCl4 (carbon tetrachloride)-induced liver injury, HFD (high fat diet)-induced NAFLD (nonalcoholic fatty liver disease) and MCD (methionine-choline deficient diet)-induced NASH (nonalcoholic steatohepatitis), no marked difference in Mirt2 expression was observed between the livers from the control and disease groups (Fig. 8c–e). As demonstrated in our in vitro experiment, LPS and IL-4, which facilitate macrophage differentiation into two extremes of polarization (inflammatory M1 and anti-inflammatory M2), increased and decreased Mirt2 expression in macrophages respectively (Fig. 2a, b, Supplementary Fig. 1c). Consistently, in pathological tissues, such as atherosclerotic plaques from ApoE−/− mice and adipose tissues from obese mice, Mirt2 was colocalized predominantly with iNOS+ M1 macrophages but was barely detected in Arg1+ M2 macrophages (Fig. 8i, j).

---

**Fig. 5** Mirt2 inhibits TRAF6 oligomerization and auto-ubiquitination. **a** Silver-stained SDS-PAGE gel analysis of proteins in macrophages that are bound to biotinylated lncRNA-Mirt2. The highlighted regions were analyzed by mass spectrometry, identifying TRAF6 as a protein unique to Mirt2. **b** Immunoblotting analysis of proteins in macrophages bound to biotinylated Mirt2 using anti-TRAF6 antibody. **c** RNA immunoprecipitation (RIP) analysis to determine the recovery of Mirt2 in macrophages using anti-TRAF6 antibody. IgG served as control. Data represent the mean ± SEM of three independent experiments. *$P < 0.05$ vs. IgG group. **d** RNA FISH technology and immunofluorescent analysis to determine the co-localization of Mirt2 (green) and TRAF6 (red) in macrophages. DAPI was used for nucleus staining (blue). The images correspond to a single slice of Z-stacks and the experiments were repeated three times. **e** RNA pull-down analysis was employed to determine the interaction of TRAF6 with full-length or truncations of Mirt2. **f** Constructs for myc-tagged TRAF6 (wild type or domain truncation mutants) were transfected into HEK293T cells, pulled down by biotinylated Mirt2 transcript and examined by Western blot with antibody to myc. Bottom: the domain structure of TRAF6. **g** Immunoprecipitation of HA-TRAF6 (myc-TRAF6) followed by immunoblot analysis of myc-TRAF6 (HA-TRAF6) in HEK293T cells expressing both TRAF6 constructs to detect TRAF6 oligomerization in the presence or absence of Mirt2. **h** Immunoblot analysis of the ubiquitination of TRAF6 in HEK293T cells transfected to express myc-tagged TRAF6 and HA-tagged ubiquitin. Left panel: Total ubiquitination of TRAF6; Middle panel: K48-linked and K63-linked ubiquitination of TRAF6; Right panel: The effects of different concentrations of Mirt2 on the ubiquitination of TRAF6. **i** Immunoblot analysis of the K63-linked ubiquitination of endogenous TRAF6 in macrophages. **j** Immunoprecipitation of HA-Ubc13 (myc-TRAF6) followed by immunoblot analysis of myc-TRAF6 (HA-Ubc13) in HEK293T cells expressing both constructs to detect the interaction of TRAF6 and Ubc13. **k** Immunoblot analysis of the content of Ubc13 immunoprecipitated using TRAF6 antibody in macrophages. **l** Effects of Mirt2 knockdown on the phosphorylation of p65 and Jnk in macrophages transfected with si-TRAF6 or scramble siRNA, as determined by Western blot

## Discussion

A large body of evidence has indicated that lncRNAs are involved in inflammatory processes[16, 29, 30]. In this study, we investigated lncRNA expression profiles in response to LPS treatment in macrophages and identified a LPS-induced upregulated lncRNA (Mirt2). Activation of the LPS-p38-Stat1 and LPS-IFN-α/β-Stat1 pathways transcriptionally promoted lncRNA Mirt2 expression, which was mainly enriched in the cytoplasm. Mirt2 effectively relieved the inflammatory responses induced by LPS through inhibition of TRAF6 oligomerization and auto-ubiquitination.

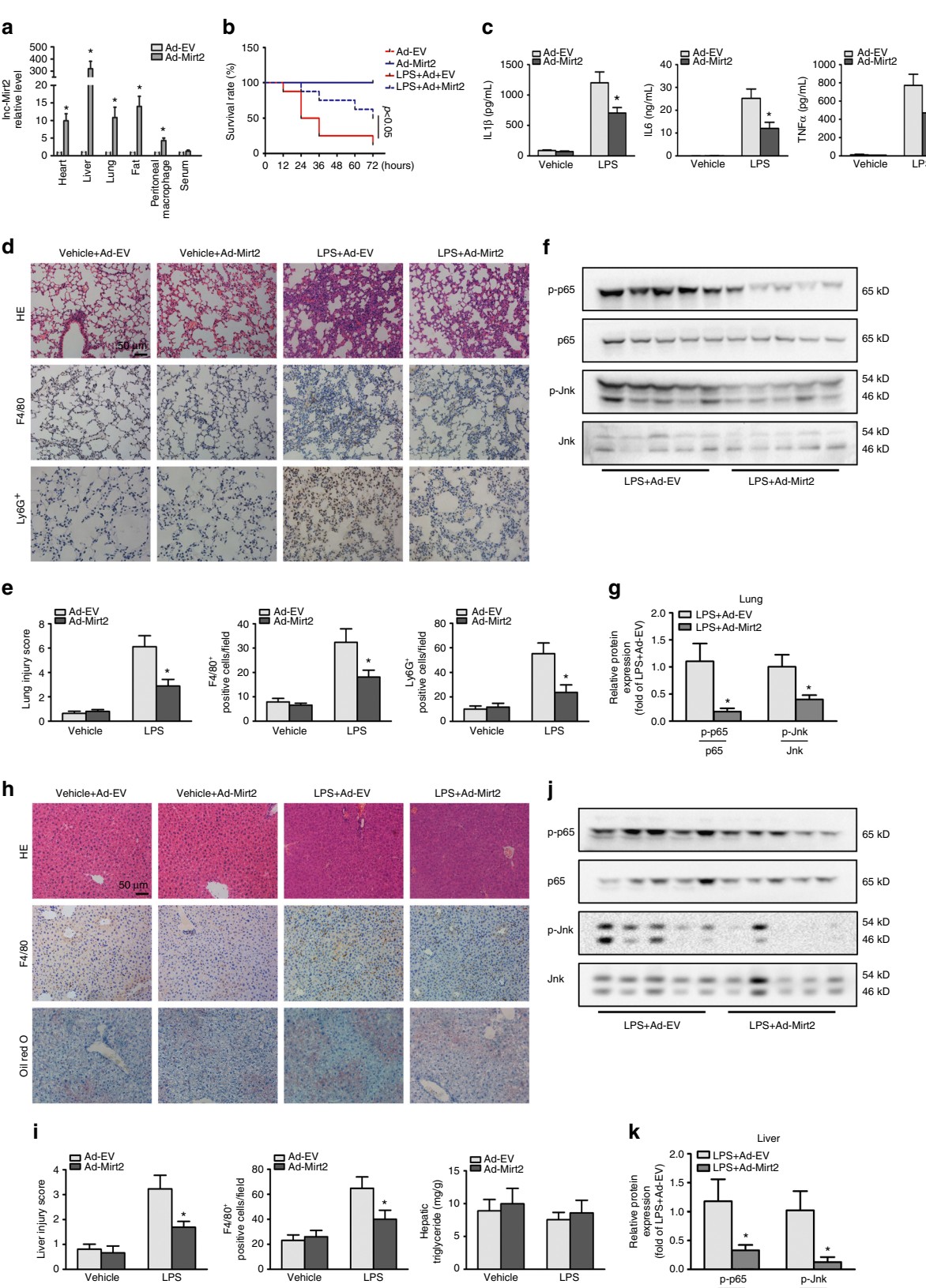

After recognizing PAMPs, TLR4 recruits specific adaptor proteins, including MyD88 and TRIF, and sequentially stimulates multiple signaling pathways to induce target gene expression[6]. Whereas the TLR4-MyD88-IRAKs-TRAF6-dependent pathway is a major mediator in inducing the expression of genes encoding proinflammatory mediators, the TLR4-TRIF-TRAF3-IRF3 (interferon regulatory factor 3)-dependent cascade is critical for the induction of genes encoding type I IFNs, as well as IFN-dependent genes (e.g., isg15), which promote the adaptation of cells to environmental conditions[7]. Our data suggest differential roles for Mirt2 in mediating TLR4 signaling, such as restricting the MyD88-dependent MAPK and NF-κB signaling cascades by inhibiting TRAF6 oligomerization and ubiquitination; conversely, no effects were observed for TRIF-dependent TRAF3 ubiquitination and the expression of type I IFNs.

K63 auto-polyubiquitination of TRAF6, which is assisted by the E2 enzyme Uev1a/Ubc13, has been shown to be essential for its activation and is counteracted by the deubiquitnases A20, CYLD, MCPIP1, USP4, and USP2a[31–36]. As demonstrated in our study, the Ring domain of TRAF6, which is responsible for the interaction with Ubc13, is also the domain involved in the interaction with Mirt2. We speculate that the interaction of Mirt2 with TRAF6 may conceal its ubiquitination sites for the E2 ubiquitin ligase Ubc13 and thus induce a decreased in the TRAF6 ubiquitination level. These effects seemed to not be mediated by an accelerated deubiuitination process, because the level of A20, which is a critical deubiquitnase, declined and CYLD remained unchanged upon Mirt2 overexpression.

In our study, the Mirt2 level in macrophages was low under resting conditions. Upon LPS stimulation, Mirt2 expression was significantly increased at 6 h and peaked at 12 h. Knockdown of Mirt2 resulted in sustained activation of inflammatory pathways at 6 and 10 h and a remarkable increase in inflammatory factors at 20 h, at which time their levels dropped rapidly in the control group. This finding led us to conclude that the recession of inflammatory responses at the late stage in LPS-activated macrophages might be attributed, at least partly to the elevated Mirt2 level. Although the Mirt2 level was elevated in LPS-stimulated macrophages, increasing Mirt2 expression using an adenovirus still exerted remarkable anti-inflammatory effects at the early and late stages. This result could be due to rapid activation of the inflammatory pathways before Mirt2 elevation upon LPS stimulation.

Classical M1 and alternative M2 macrophages, which are induced by LPS and IL-4, respectively, represent two extremes of a dynamic changing state of macrophage activation. M1-M2 polarization of macrophages is a tightly controlled process that entails a set of signaling pathways and transcriptional and post-transcriptional regulatory networks[37, 38]. M1 macrophages exhibit pro-inflammatory activities that are necessary for host defense, and M2 macrophages are involved in the tissue repair required for the restoration of homeostasis[39, 40]. In contrast to LPS, IL-4 stimulation through the Jak-Stat6 signaling pathway resulted in a rapid decrease in the Mirt2 level that was significant at 2 h and most obvious at 6 h. Restoring Mirt2 expression promoted IL-4-induced M2 polarization with a more rapid and remarkable increase in the levels of M2 markers, such as Arg1, CD206, Fizz1, Ym1, and CCL24. Unlike most inflammatory factors, which peaked at 4 h and subsequently dropped upon LPS stimulation, the M2 markers induced by IL-4 increased gradually and peaked at 20 h. This scenario suggests that a rapid decline in Mirt2 may have led to delayed M2 polarization. Thus, the LPS-induced increase of Mirt2 restricted the sustained and excessive activation of inflammatory responses at the late stage, and IL-4-induced a decline in Mirt2 that restrained macrophage M2 polarization at the early stage. For the latter possibility, the physiological roles and molecular mechanisms have not been elaborated. The correlation between the Mirt2 level and macrophage polarization is depicted in Supplementary Fig. 7.

Notably, the ability of Mirt2 to regulate cellular inflammatory responses is not limited to myeloid cells. The expression of inflammatory factors induced by LPS could also be inhibited by Mirt2, at least in cultured tracheal epithelial cells and hepatocytes,. However, we did not observe similar effects in adipose cells, cardiomyocytes, and cardiac fibroblasts. This discrepancy is probably attributed to the differences in physiological characteristics and regulatory mechanisms on inflammatory processes in diverse cell types. Moreover, in our in vivo animal experiment, adenovirus-mediated gene transfer of Mirt2 definitely relieved LPS-induced endotoxemia and organ dysfunction, especially in the lung and liver. However, these remarkable protective effects of Mirt2 seemed to not be exclusively dependent on macrophage inhibition, since systemic adenovirus gene transfer actually affected various tissues, and the infection efficiency was more limited for myeloid cells than for parenchymal cells such as hepatocytes and cardiomyocytes. To further elucidate the effects of Mirt2 on specific cell types of certain tissues during endotoxemia, cell type-selective genetic manipulation should be taken into account in our future studies.

Notably, we identified that the Mirt2 level was dramatically increased in the plasma of mice with endotoxemia. In light of our results, we propose that this release may be attributable to the increased Mirt2 levels in the diseased organs. However, adenovirus-mediated gene transfer did not lead to a further increase in the plasma Mirt2 level. This finding may suggest that the release of Mirt2 is under stringent control and that simply

**Fig. 6** Mirt2 attenuates LPS-induced endotoxemia. Endotoxemia was induced in C57BL/6 mice by intraperitoneal injection of LPS (25 mg/kg), and control animals were administered with equivalent volumes of normal saline. Adenovirus (Ad-Mirt2 or Ad-EV) were delivered into mice by tail veil injection 3 days before LPS challenge. **a** Adenovirus infection efficiency at 72 h after adenovirus administration. **b** Survival curve of mice with endotoxemia (n = 12). **c** Levels of cytokines (IL-1-β, IL-6 and TNF) in serum of mice challenged with LPS for 6 h. Data are expressed as mean ± SEM (n = 8). *P < 0.05 vs. Ad-EV group. **d** Histopathology in lung of Ad-EV or Ad-Mirt2 treated mice 24 h after LPS challenge. Upper panel: Hematoxylin and eosin staining; Middle panel: Immunohistochemical staining for macrophage marker F4/80; Lower panel: Immunohistochemical staining for neutrophil marker Ly6G. **e** Quantitative analysis of lung injury in panel **d**. Left panel: Lung injury score; Middle panel: Quantitative analysis of F4/80 positive cells; Right panel: Quantitative analysis of Ly6G positive cells. Data are expressed as mean ± SEM (n = 8). *P < 0.05 vs. Ad-EV group. **f** Western blot analysis for the phosphorylation of p65 and Jnk in lung of endotoxemia mice. **g** Quantification of band density in panel **f**. Data are expressed as mean ± SEM (n = 5). *P < 0.05 vs. Ad-EV group. **h** Histopathology in liver of Ad-EV or Ad-Mirt2 treated mice 24 h after LPS challenge. Upper panel: Hematoxylin and eosin staining; Middle panel: Immunohistochemical staining for macrophage marker F4/80; Lower panel: Oil Red O staining. **i** Quantitative analysis of liver injury in panel h. Left panel: Liver injury score; Middle panel: Quantitative analysis of F4/80 positive cells; Right panel: Determination of the contents of triglyceride. Data are expressed as mean ± SEM (n = 8). *P < 0.05 vs. Ad-EV group. **j** Western blot analysis for the phosphorylation of p65 and Jnk in liver of endotoxemia mice. **k** Quantification of band density in panel **j**. Data are expressed as mean ± SEM (n = 5). *P < 0.05 vs. Ad-EV group. Two-tailed Student's t-test for two groups

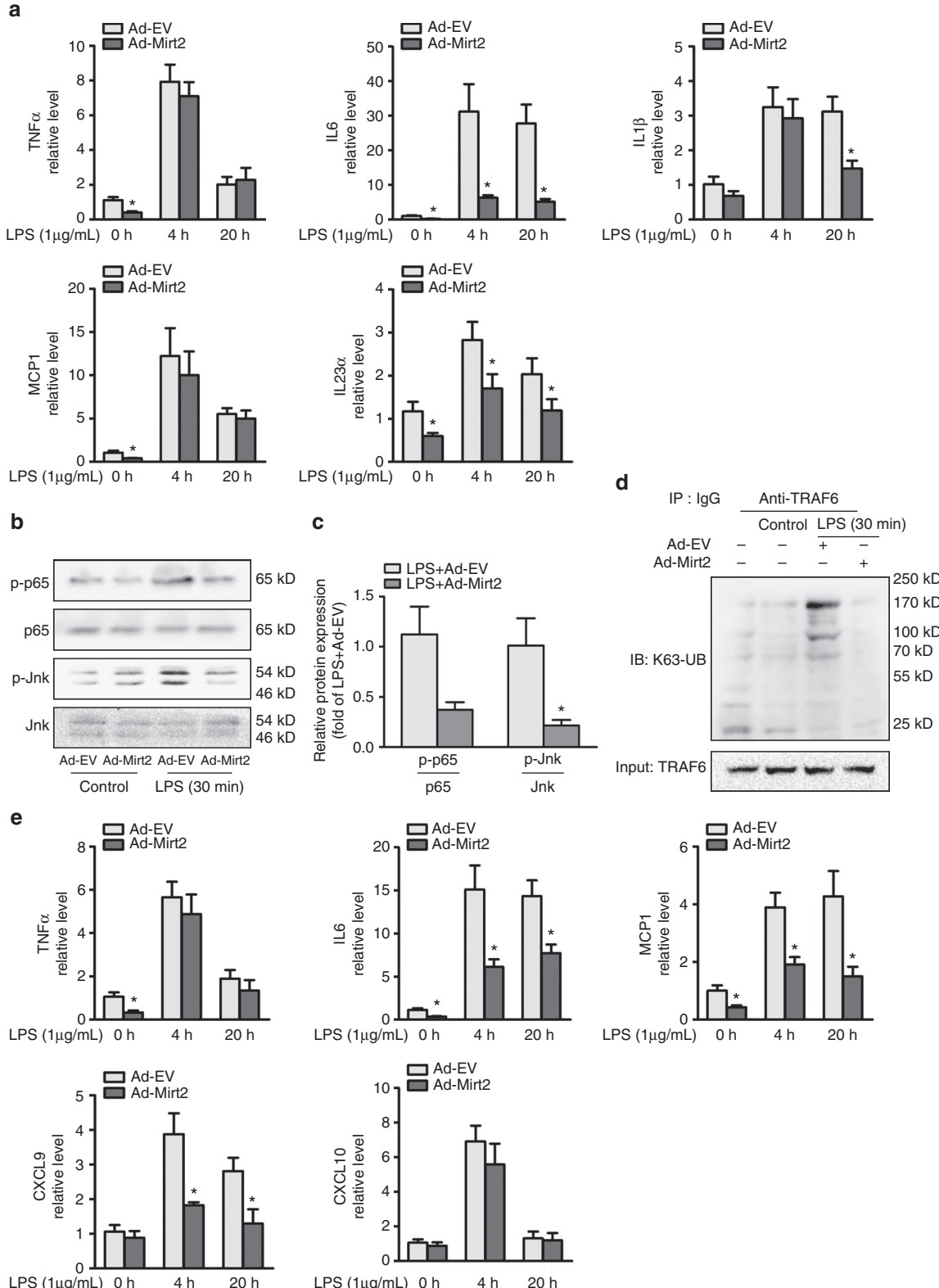

**Fig. 7** Mirt2 has anti-inflammatory effects in tracheal epithelial cells and hepatocytes. **a** Effects of Mirt2 overexpression on the expression of inflammatory factors in tracheal epithelial cells, as determined by qRT–PCR. **b** Effects of Mirt2 overexpression on the phosphorylation of p65 and Jnk in tracheal epithelial cells, as determined by Western blot. **c** Quantification of band density in panel **b**. **d** Immunoblot analysis of the K63-linked ubiquitination of endogenous TRAF6 in Mirt2-overexpression tracheal epithelial cells treated with LPS, assessed after immunoprecipitation with TRAF6 antibody. **e** Effects of Mirt2 overexpression on the expression of inflammatory factors in hepatocytes, as determined by qRT–PCR. Data represent the mean ± SEM of three independent experiments. *$P < 0.05$ vs. Ad-EV group. Two-tailed Student's $t$-test for two groups

increasing Mirt2 expression via an adenovirus approach is not sufficient to induce the additional release of this lncRNA from the cells. Whether plasma Mirt2 is simply a by-product of the increased intracellular levels or is functionally active in disease pathology remains unclear and needs further study.

In addition, we conducted detailed analyses of human/mouse homology of Mirt2 by analyzing human RNA-seq databases for potential transcripts but could not identify a clear human homolog for this lncRNA. Surprisingly, an adenovirus carrying a murine origin Mirt2 gene also exerted significant anti-

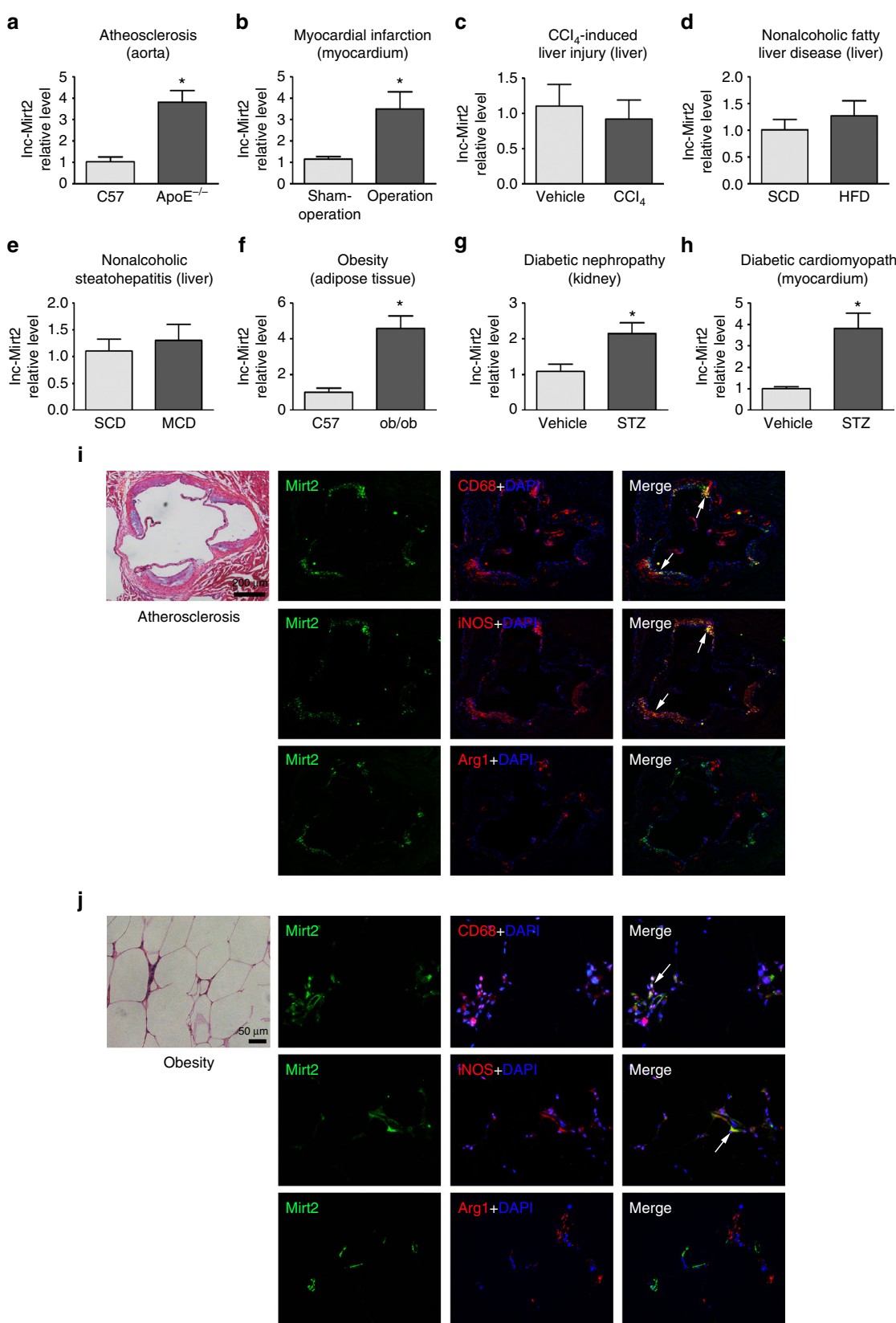

inflammatory effects in some cell types derived from humans, such as monocyte-derived macrophages and hepatocytes. Therefore, exploring whether an undiscovered human homolog of Mirt2 remains conserved in its primary sequence, structure or functions is important. This study indicates great value for Mirt2 as a potential therapeutic target of inflammatory diseases. In

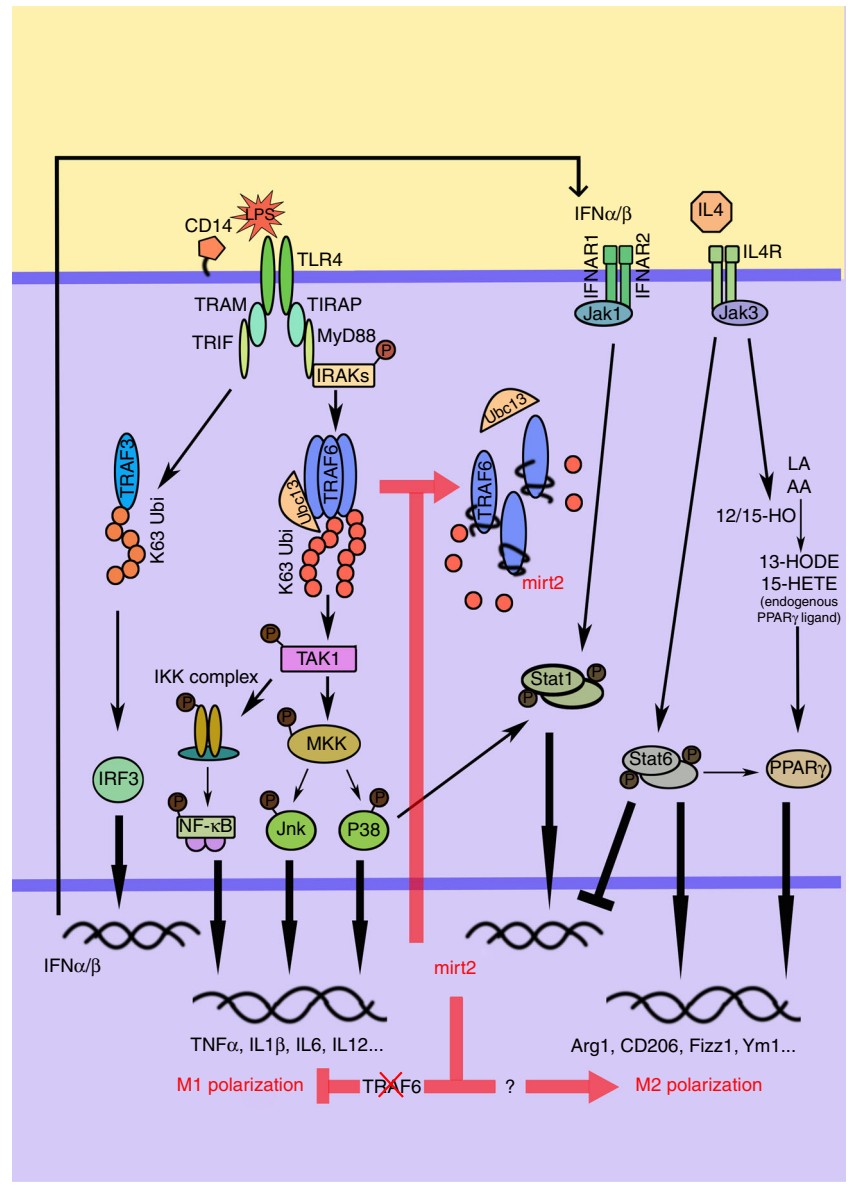

**Fig. 9** Schematic of the molecular mechanisms of Mirt2 in inflammatory responses. Activation of LPS-p38-Stat1 and LPS-IFN-α/β-Stat1 pathways transcriptionally promotes the expression of lncRNA Mirt2, and Mirt2 specifically inhibits the K63-ubiquitination of TRAF6, thus alleviating inflammatory responses after TLR4 activation and balancing macrophage polarization

**Fig. 8** Mirt2 in other inflammatory diseases. **a** Male C57BL/6J and ApoE−/− mice were fed a Western diet from 6 to 18 weeks old, then the mice were sacrificed and Mirt2 in aorta was detected. **b** Male C57BL/6J mice aged 8 weeks were suffered left-anterior descending coronary artery (LAD) ligation, and sham group as control. 24 h after operation, mice were sacrificed and Mirt2 in myocardium was detected. **c** Male C57BL/6J mice aged 12 weeks were injected intraperitoneally with a single dose of CCl4 (1 ml/kg) or an equal volume of vehicle. After 72 h, mice were sacrificed and Mirt2 in liver was detected. **d** Male C57BL/6J mice aged 6 weeks were fed with standard chow diet (SCD) or high fat diet (HFD) containing 60% calories as fat for 12 weeks. Then mice were sacrificed and Mirt2 in liver was detected. **e** Male C57BL/6J mice aged 10 weeks were fed with standard chow diet (SCD) or methionine-choline deficient diet (MCD) diet for 6 weeks. Then mice were sacrificed and Mirt2 in liver was detected. **f** Male C57BL/6J and ob/ob−/− mice were sacrificed at 24 weeks of age and Mirt2 in adipose tissue was detected. **g, h** Male C57BL/6J mice aged 10 weeks were injected intraperitoneally with 50 mg/kg/day STZ for five consecutive days. Age-matched control mice received an equal volume of vehicle. One week after STZ injection, fasting blood glucose was checked, and mice with blood glucose >300 mg/dL were used for experiments. The mice were sacrificed at 24 weeks of age and Mirt2 in kidney and myocardium was detected. **i, j** RNA FISH analysis of Mirt2 in atherosclerotic plaques from ApoE−/− mice **i**, and in adipose tissues from obese mice **j**. For colocalization analysis, sections were co-stained for Mirt2 (green) and CD68 (red, macrophage marker), iNOS (red, M1 marker) or Arg1 (red, M2 marker). DAPI was used for nucleus staining (blue). Arrows indicate Mirt2 and cell specific marker double positive cells. Data are expressed as mean ± SEM (n = 12). *P < 0.05. Two-tailed Student's t-test for two groups

summary, our study provides new insights into the dynamic and reversible nature of ubiquitin modification in the innate immune response. Mirt2, which is a LPS-induced lncRNA in murine macrophages, specifically inhibits the K63-ubiquitination of TRAF6 and thus alleviates inflammatory responses after TLR4 activation and balances macrophage polarization (Fig. 9).

## Methods

**LncRNA microarray analysis.** Total RNA was extracted from six samples (three control and three LPS treated subjects) and purified by the RNeasy Mini Kit (Qiagen, GmBH, Hilden, Germany) according to the manufacturer's recommendation. Purified total RNA was quantified using the NanoDrop ND-1000 spectrophotometer (Nano-Drop Technologies Wilmington, DE) and RNA integrity was assessed using standard denaturing agarose gel electrophoresis. LncRNA expression profiles were investigated using SBC Mouse (4 × 180 K) lncRNA Microarray version 6.0. Briefly, the RNA samples were first reverse transcribed into cDNA, and these cDNA samples were then labeled using a Low Input Quick-Amp Labeling Kit (Agilent Technologies, Santa Clara, CA, USA). After purification with the RNeasy Mini Kit, hybridization was performed using a Gene Expression Hybridization Kit in a Hybridization Oven (Agilent Technology, Santa Clara, CA, USA). The hybridized arrays were washed, fixed, and scanned using the Agilent Microarray Scanner (Agilent Technology, Santa Clara, CA, USA). Array images were analyzed and raw data were extracted using the Agilent Feature Extraction software, and Gene-Spring software was employed to finish the basic analysis of the raw data. Differentially expressed lncRNAs between the two groups were then identified through fold change, and adjusted p-value (q-value) was calculated using multiple tests. Statistically significant differential expression of lncRNA was displayed through volcano plot filtering with a threshold of fold change $\geq 2$ and $q < 0.05$. Then these differentially expressed lncRNAs were explored by using more stringent criteria (Student's t test, $P < 0.01$, fold change > 20) and filtered according to transcript abundance.

**RNA extraction and qRT-PCR.** Total RNA was extracted from cells or tissues with the use of TRIzol reagent (D9108A, Takara Bio). RNA was reverse-transcribed using the RNA PCR Kit (RR036A, Takara Bio). Quantitative polymerase chain reaction (PCR) amplification was performed with an ABI PRISM 7900 Sequence Detector system (Applied Biosystem, Foster City, CA), according to the manufacturer's instructions. Relative gene expression (Mirt2 and other host inflammatory genes, normalized to endogenous control gene β-actin) was calculated using the comparative Ct method formula $2^{-\Delta\Delta Ct}$. The real-time PCR primer sequences are shown in Supplementary Table 1.

**Western blot.** Cells or tissues were harvested at indicated times and homogenized in ice-cold suspension buffer supplemented with a proteinase inhibitor cocktail (Sigma-Aldrich). Protein concentrations were determined using the BCA Protein assay kit (Thermo Scientific, Waltham, MA). Equal amounts of protein were fractionated by SDS polyacrylamide gels, followed by immunoblotting with the following primary antibodies: NF-κB Pathway Sampler Kit (diluted at 1:1000, 9936, CST, MA), phospho-MAPK Family Antibody Sampler Kit (diluted at 1:1000, 9910, CST, MA), MAPK Family Antibody Sampler Kit (diluted at 1:1000, 9926, CST, MA), phospho-Stat6 antibody (Rabbit polyclonal, diluted at 1:1000, ab28829, Abcam), Stat6 antibody (Rabbit polyclonal, diluted at 1:1000, ab44718, Abcam), TRAF6 antibody (Rabbit polyclonal, diluted at 1:1000, ab94720, Abcam), TLR4 antibody (Rabbit polyclonal, diluted at 1:1000, ab13556, Abcam), CD14 antibody (Rabbit polyclonal, diluted at 1:1000, ab203294, Abcam), Myd88 antibody (Rabbit polyclonal, diluted at 1:1000, ab2064, Abcam), PPARγ antibody (Rabbit monoclonal (C26H12), diluted at 1:1000, sc-7273, Santa Cruz Biotechnology). Membranes were then incubated with peroxidase-conjugated secondary antibody, and specific bands were detected with a Bio-Rad (Hercules, CA) imaging system. Original blots were provided in Supplementary Fig. 8.

**RNA FISH and protein immunofluorescence.** Cells grown on glass coverslips or tissue slices were fixed in 4% paraformaldehyde for 30 min and permeabilized with 0.1% Triton X-100. Then cells were washed three times with phosphate buffer saline (PBS) and treated with pre-hybridization buffer (2 × saline sodium citrate, 10% formamide). The FISH (Fluorescence in situ hybridization) probes to Mirt2 (Exiqon) were resuspended in hybridization buffer (2 × saline sodium citrate, 10% formamide, 10% dextran sulfate) to a final concentration of 250 nM per probe set. Hybridization was carried out in a humidified chamber at 37 °C for 16 h. After incubated with the Mirt2 probe, cells were washed three times with PBS and treated with TRAF6 antibody at 37 °C; 2 h later, cells were washed three times with PBS and treated with fluorescent-labeled secondary antibody for 1 h at 37 °C. We continued with several rounds of washing (which included an optional 4′,6-diamidino-2-phenylindole staining step) and finished with mounting the coverslip onto a microscope slide using an anti-fade mounting medium. The information about antibodies used for imaging and the secondary antibodies is as follows: TRAF6 antibody (Rabbit polyclonal, diluted at 1:200, ab94720, Abcam), p65 antibody (Rabbit monoclonal (E379), diluted at 1:100, ab32536, Abcam), CD68

antibody (Mouse monoclonal (C68/684), diluted at 1:200, ab201340, Abcam), iNOS antibody (Rabbit monoclonal (EPR16635), diluted at 1:100, ab178945, Abcam), Arginase antibody (Rabbit polyclonal, diluted at 1:1000, ab91279, Abcam), Goat anti-Mouse IgG (ab150115, Abcam), Goat anti-Rabbit IgG (ab150079, Abcam).

**RNA pull-down and mass spectrometry.** Biotin-labeled RNAs were in vitro transcribed using the Biotin RNA Labeling Mix and T7 RNA polymerase (Ambion), and purified with the RNeasy Mini Kit (QIAGEN) on-column digestion of DNA. The biotinylated Mirt2, antisense Mirt2, or truncated Mirt2 was incubated with cell lysates (containing Rnasin) overnight at 4 °C. The interacting complexes were purified with streptavidin beads for 3 h at room temperature, and visualized by silver-staining (Pierce silver stain kit, Thermo Scientific) for mass spectrometry or by immunoblotting using specific antibody to TRAF6 or TAK1. For Mass Spectrometry, a specific band present in the experimental lane was extracted (corresponding region in the control lane was also extracted) for further analysis.

**RNA immunoprecipitation.** Cells were treated with 0.3% formaldehyde for 10 min at 37 °C. In total 1.25 M glycine dissolved in PBS was added to a concentration of 0.125 M, and the sample was incubated for 5 min at room temperature. Then cells were washed twice in PBS and pelleted. The pellet was resuspended in RIPA buffer (50 mM Tris, pH 7.4, 150 mM NaCl, 1 mM EDTA, 0.1% SDS, 1% NP-40, 0.5% sodium deoxycholate, 0.5 mM DTT and 1 mM PMSF/cocktail)[41], incubated on ice with frequent vortex for 30 min. TRAF6 antibody or IgG were added and samples were incubated overnight at 4 °C. The RNA/protein complex was recovered with protein G Dynabeads and washed with RIPA buffer several times. RNA was recovered with Trizol and analyzed by RT–PCR.

**Plasmid constructs and reporter assay.** The wild-type TRAF6 plasmid was obtained from ORIGENE (MR208489), and the truncated constructs of TRAF6 (1–259aa, 1–357aa, 260–530aa, 358–530aa, 1–149aa, 150–259aa) were prepared by PCR and cloned into pCMV6-Entry mammalian vector with C-terminal Myc tag. For reporter assay, Mirt2 promoters with different lengths were PCR amplified from mouse genomic DNA and cloned into the pGL3-Basic vector (Promega, Madison, WI) using the One Step Cloning Kit (C112-02, Vazyme Biotech Ltd., Nanjing, China). Luciferase reporter constructs were cotransfected with an internal control plasmid, pRL-TK (Renilla luciferase reporter plasmid, Promega), into RAW264.7 cells, followed by stimulation with LPS or IL-4. Then, cells were harvested, lysed, and the luciferase activity was determined with the Dual Luciferase Reporter Assay Kit (Promega), according to the manufacturer's instruction.

**Ubiquitination assays.** The expression constructs wild type UB-HA, K48-UB-HA (all lysines except lysine 48 mutated to arginines) and K63-UB-HA (all lysines except lysine 63 mutated to arginines) were a gift from Dr Hongliang Li (Wuhan University, Wuhan, China). For in vivo ubiquitination analysis via immunoprecipitation, cells were lysed in SDS lysis buffer (20 mM Tris–HCl, pH 7.4, 150 mM NaCl, 1 mM EDTA, 1% SDS) containing Protease Inhibitor Cocktail Tablets (04693132001, Roche) and denatured by heating for 5 min. The lysates were diluted to 0.1% SDS by RIPA buffer and centrifuged at 13,200 rpm at 4 °C for 10 min. The supernatants were subjected to immunoprecipitation with the indicated antibodies and protein A/G-Sepharose (GE healthcare) for 3 h, at 4 °C. After three times washing with RIPA buffer, bound proteins were separated with SDS-PAGE and analyzed by Western blot with antibodies specific for ubiquitin. Uncropped full-length blots are provided in Supplementary Fig. 8.

**Generation of recombinant adenovirus.** Replication-defective recombinant adenovirus carrying the entire coding sequence of Mirt2 (Ad-Mirt2) was constructed with the Adenovirus Expression Vector Kit (Takara Bio Inc., Kusatsu, Japan). An adenovirus-only-containing green fluorescence protein (GFP) was used as a negative control (Ad-EV). To generate adenovirus expressing shRNA against Mirt2 (Ad-sh-Mirt2), 3 siRNAs for mouse Mirt2 were designed and the one with the optimal knockdown efficiency was chosen to create shRNA and then recombined into adenoviral vectors. The target sequence is as follows: CCTCCTCGACG-GATTTCAA. The negative control adenovirus was designed to express non-targeting "universal control" shRNA (Ad-shNC). Amplification and purification of recombinant adenovirus was performed according to the manufacturer's instructions (Takara Bio).

**Mice.** Male C57BL/6 mice aged 8–12 weeks were bred and maintained under conventional housing conditions in our animal facility, and all the animal experimental protocols were approved by the Ethics Committee of Union Hospital, Huazhong University of Science and Technology, China, and were conducted in accordance with the National Institutes of Health (NIH) Guide for the Care and Use of Laboratory Animals. Adenovirus (Ad-Mirt2 or Ad-EV, 5 × 10^9 pfu/mouse) were delivered into mice by tail veil injection, 3 days before endotoxemia model established. For the LPS-induced endotoxemia model, mice were injected intraperitoneally with LPS (Escherichia coli 0111:B4, Sigma-Aldrich, St. Louis, MO) at a dose of 25 mg/kg. Control animals were administered with equivalent volumes of

normal saline. Their survival was monitored and recorded until 72 h. All mice were killed at 30 min for detection of activation of inflammatory pathways, at 6 h for determination of inflammatory factors in plasma, and at 24 h for histological evaluation. Animal data were excluded from experiments based on pre-established criteria of visible abnormal tissue structure during sample harvest or other health issues including fighting wounds.

**ELISA.** The concentration of TNF, IL-6, and IL-1β in plasma from animals was calculated by using ELISA kits according to the manufacturer's instructions. ELISA kits were: TNF (MTA00B, R&D Systems), IL-6 (M6000B, R&D Systems), IL-1β (MLB00C, R&D Systems).

**Cell culture.** Peritoneal macrophages were isolated from C57BL/6 mice[42]. Briefly, the peritoneal cavity was first lavaged with sterile NaCl (0.9%), and then the lavage fluids were collected, pooled and centrifuged. Cell pellets were suspended in RPMI1640 medium (Gibco) supplemented with 10% fetal bovine serum (Gibco). Macrophages were allowed to adhere in culture plates for 2 h. Non-adherent cells were removed by washing and the adherent cells were maintained for 24 h in 10% serum-containing medium for further study. Primary tracheal epithelial cells, hepatocytes, adipocytes, cardiomyocytes and cardiac fibroblasts were prepared, as previously described[43–46]. HEK293T cells (CRL-11268) and RAW264.7 cells (TIB-71) were obtained from ATCC and cultured in complete DMEM medium supplemented with 10% fetal bovine serum for further study. All of the cell lines were free of mycoplasma contamination (tested by the vendors using the MycoAlert kit from Lonza). No cell lines used in this study are found in the database of commonly misidentified cell lines (ICLAC and NCBI Biosample) based on short tandem repeats (STR) profiling performed by vendors.

**Statistical analysis.** GraphPad Prism software (GraphPad Software Inc., La Jolla, CA) was used for statistical analyses. Band intensity in western blot images was quantified with Image J Software. Values are expressed as means ± SEM of at least three independent experiments. Student's $t$ test was used to assess the statistical significance of the differences between two groups. For multiple groups, significance was evaluated by one-way ANOVA with Bonferroni test. $P < 0.05$ was considered statistically significant. For animal survival analysis, the Kaplan–Meier method was empolyed to generate graphs, and the survival curves were analyzed using log-rank analysis. Randomization and blinding strategy was used whenever possible. Animal cohort sizes were determined on the basis of similar previous studies.

**Data availability.** The data that support the findings of this study are available from the corresponding author upon request.

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

## Acknowledgements

This work was supported by the grants from the National Natural Science Foundation of China (No. 81570405) and National Key Research and Development Program of China (No. 2016YFA0101101).

## Author contributions

K.H. and M.D. designed the experiments and wrote the manuscript. M.D., L.Y., X.T., D.H., X.W., Z.Z., X.M., X.L., and L.Y. performed the experiments. M.D., K.H., K.H., and F.Z. analyzed the data. Y.W., X.L., and D.H. contributed reagents/materials/analysis tools.

## Additional information

**Competing interests:** The authors declare no competing financial interests.

