## [Peer review file · Nature Communications]

Reviewers' comments:

Reviewer #1 (lncRNAs)(Remarks to the Author):

In the manuscript entitled "A long non-coding RNA Mirt2 induced by TLRs mediates repression of inflammatory responses", Du et al. demonstrate that the lncRNA, Mirt2, is induced by TLR4 activation with LPS. Moreover, the authors suggest that mirt2 is a regulator of inflammation through disruption of TRAF6-mediated signaling. The authors suggest that this is due to direct RNA-protein interactions between Mirt2 and TRAF6, inhibiting K63-Ub and thus downstream inflammatory gene activation. Moreover, the authors go on to demonstrate that in vivo overexpression of Mirt2 protects mice from the unabated inflammatory response induced by LPS. Given the lack of knowledge on the regulation and function of Mirt2, this article is of great interest to the field. However, while the data is compelling, due to omission of the experimental details and data quality, the manuscript cannot be accurately reviewed as presented.

In the manuscript entitled "A long non-coding RNA Mirt2 induced by TLRs mediates repression of inflammatory responses", Du et al. demonstrate that the lncRNA, Mirt2, is induced by TLR4 activation with LPS. Moreover, the authors suggest that mirt2 is a regulator of inflammation through disruption of TRAF6-mediated signaling. The authors suggest that this is due to direct RNA-protein interactions between Mirt2 and TRAF6, inhibiting K63-Ub and thus downstream inflammatory gene activation. Moreover, the authors go on to demonstrate that in vivo overexpression of Mirt2 protects mice from the unabated inflammatory response induced by LPS. Given the lack of knowledge on the regulation and function of Mirt2, this article is of great interest to the field. However, while the data is compelling, due to omission of the experimental details and data quality, the manuscript cannot be accurately reviewed as presented.

Major points:

1) In figure 1, where the p-values derived from the microarray data adjusted for multiple testing? If not, does the significance of differentially expressed genes change? Additionally, it would be beneficial to avoid the use of greenred heatmaps and plots when representing these data. In the volcano plot, how many genes are being represented? Of those, how many of these genes were found to be significantly upregulated/downregulated? Perhaps the authors could consider color coding lncRNAs to distinguish them from coding and other non-coding transcripts as it makes it unclear to visualize mirt2 as the most highly abundant linc.

2) Is the induction of Mirt2 by LPS dose-dependent? The concentrations used for LPS stimulation are quite high. Would the authors expect to see this induction at a lower dose? Moreover, the authors should indicate the concentration of TLR agonists used to assess the induction of Mirt2. According to the Materials and Methods, the authors utilized a student's t-test to evaluate statistical significance, which is not an appropriate statistical test when comparing more than 2 groups.

3) In figure 2, the authors should clarify the normalization method for their qPCR data. What reference genes were used for the comparison of relative expression of Mirt2 and other host inflammatory genes? Furthermore, the immunofluorescence images (Fig.2C and E) are too small and low in resolution. Perhaps the authors should zoom in on the cells and provide a quantification of the number of positive cells in the field. Additionally, the authors should clarify the number of replicates used on their FISH experiments.

4) Please indicate the concentration of chemical inhibitors used in this study.

5) In Fig. 3, the authors demonstrate that p38 is crucial for the induction of Mirt2 by showing that chemical inhibition leads to decreased Mirt2 expression. However, has also been shown that p38 is important for the induction of Ifnb in macrophages. Thus, the authors should explore whether this

effect is due to decreased *Ifnb* expression after inhibitor treatment. On the other hand, the authors should show whether or not other TLR stimulations led to robust induction of *Ifnb*. One would expect that a robust activation of TLR3 would have led to the secretion of type I IFNs and thus activated the phosphorylation of STAT1 and thereby *Mirt2* induction.

6) While the authors claim that there are no changes in the expression of TLR4 and/or other related adaptor molecules following silencing or overexpression of *Mirt2*, the data as presented (SFig.3 F-H) is hard to interpret due to contrast/exposure issues.

7) The authors should provide better quality WB images for figure 3. The images presented are mostly underexposed. Additionally, the p65 nuclear translocation figure is too small to observe changes in the relative amount of translocation. Perhaps the authors could show a representative zoomed in image and quantify the number of events where there is decreased translocation. As mentioned before, the authors should provide details as to the number of replicates these images represent.

8) In figure 5, the authors should avoid the use of increased contrast or underexposure of blots in order to accurately review the data. Additionally, do the authors see colocalization between *Mirt2* and TRAF6 in non-stimulated macrophages? Is the endogenous interaction dependent of stimulation? Please indicate whether the images in Fig. 5D correspond to a single slice of Z-stacks or MFI representations. Also indicate the number of field/replicates used to determine these interactions. Given the high concentration of LPS utilized in this study, the authors should include a housekeeping gene to demonstrate LPS-treated cells are viable.

9) Does *Mirt2* expression affect the signal transduction of other receptors upstream of TRAF6?

10) In line 576, the authors acknowledge that there is no human homologue of *Mirt2*. However, they indicate that murine-derived *Mirt2* might have anti-inflammatory in human cells. The authors should include this data as *Mirt2* might be have therapeutic potential in the regulation of inflammation.

11) In SFig.3, it seems that the phosphorylation of TAK1 is diminished after overexpression of *Mirt2*. Have the authors explored the interaction between TAK1 and *Mirt2*? One would expect that the Ub of TRAF6 would be impacted.

12) KD efficiencies for all the siRNAs should be shown by western blot.

Minor points:

1) The manuscript needs to be revised for English grammar and clarity. Please change *mirt2* to *Mirt2*.

2) If possible, please use bigger fonts and enlarge the figures as it is hard to evaluate the data and read the legends. This is especially important for the evaluation of microscopy/histology data.

3) Please include in the Materials and Methods information regarding the antibodies used for imaging and the secondary antibodies used. Please indicate the amino acid residues evaluated for phosphorylation.

Reviewer #2 (TRAF6 signalling)(Remarks to the Author):

The manuscript from Du et al have identified mirt2, a long non-coding RNA as a negative regulator of inflammatory responses, via its inhibitory effects on TRAF6. By searching for LPS-induced long non-coding RNA the authors have identified mirt2 as a target, which is expressed mainly the cytoplasm in activated macrophages from septic mice. They provide also experimental evidence for that p38-Stat1 and IFN-Stat1 signaling pathways regulates expression of mirt2. Adenovirus-induced overexpression of mirt2 in mice, suppress LPS-induced inflammatory responses in macrophages. Interestingly, mirt2 associates to the Ring-domain and Zn-fingers in TRAF6 and can thereby prevent the catalytic activity of TRAF6. In detailed in vivo experiments performed in mice, the authors provide detailed and convincing data for a physiological role of mirt2 to regulate inflammatory responses and polarization of macrophages. Moreover and interestingly, mirt2 is found to be specifically expressed in M1 macrophages in pathological conditions such as atherosclerosis, cardiac infarction, diabetic nephropathy and obesity.

The experimental design is very good and the presented data are convincing and provides a molecular understanding for the regulation of mirt2, its role in sepsis and pathological conditions and its inhibitory effects on TRAF6. Why the kinetics for regulation of mirt2 is relatively slow is interesting but could represent a physiological response to counteract acute inflammatory responses.

Minor comments:

Figure 4h, nuclear p65 should be quantified.

Figure 7d, the quality of this figure could be improved.

Figure 9. The schematic diagram is good for the reader to more easily access the presented data. The big red arrow for mirt2 in it's center could maybe be reduced in size and the long bended arrow in the left for IFN alpha/beta could be straight lines as well. PPARgamma should stand together.

Main text: There are some minor mistakes in the main text on line 102,435, 541 and 542. Please check.

Statistical analyses: The authors could consider to use other statistical analyses such as ANOVA, to strengthen their report.

Reviewer #3 (Sepsis and TLR4 signalling)(Remarks to the Author):

1. The definition of sepsis is infection with organ dysfunction due to a dysregulated immune response. Treating animals with LPS is not sepsis but instead is endotoxemia. All reference to sepsis should be removed from the papaer and the figures when LPS is used.
2. The results suggest that TLR-induced type 1 IFN drives Mirt2 expression. Is this MyD88 or TRIF-dependent? This is important because the induction by some of the agonists would be expected to drive signaling through MyD88 and not TRIF; however, TRIF signaling is more prominent in type 1 interferon expression.
3. In figure 2E, there is an impressive elevation of Mirt2 in the serum. What is the meaning of this? Can systemic administration of Mirt2 lead to suppression of TLR4 signaling? Is Mirt2 an intercellular signaling molecule? Understanding this would elevate the significance of the observation and add to the paper.
4. In figure 4, the authors should provide evidence that Mirt is either over or under expressed by the adenoviral vectors.
5. In figure 4H, there does not appear to be any difference induced by Mirt2 transduction in P65 translocation.
6. In figure 6A, the authors need to give the time point for the measurement of Mirt2. It would be

important to be sure to have measured Mirt2 expression at 72 hours, the time point that LPS was given. Why wait 72 hours?

7. What is the circulating levels of Mirt2 after the AdMirt2 transduction?

8. It would be important to remove cells such as macrophages from the animals receiving the AdMirt2 and test their responsiveness in vitro to assure that Mirt2 transduction blocked LPS responsiveness in one of the key target cells.

9. It would be helpful for the authors to provide the concentrations of the LPS used and the time points studied in the text.

10. In figure 8, the time point of Mirt2 measurement relevant to the age of the mice or the time in the course of the disease process needs to be provided.

Response to reviewers' comments

Reviewer #1 (lncRNAs) (Remarks to the Author):

In the manuscript entitled “A long non-coding RNA Mirt2 induced by TLRs mediates repression of inflammatory responses”, Du et al. demonstrate that the lncRNA, Mirt2, is induced by TLR4 activation with LPS. Moreover, the authors suggest that mirt2 is a regulator of inflammation through disruption of TRAF6-mediated signaling. The authors suggest that this is due to direct RNA-protein interactions between Mirt2 and TRAF6, inhibiting K63-Ub and thus downstream inflammatory gene activation. Moreover, the authors go on to demonstrate that in vivo overexpression of Mirt2 protects mice from the unabated inflammatory response induced by LPS. Given the lack of knowledge on the regulation and function of Mirt2, this article is of great interest to the field. However, while the data is compelling, due to omission of the experimental details and data quality, the manuscript cannot be accurately reviewed as presented.

In the manuscript entitled “A long non-coding RNA Mirt2 induced by TLRs mediates repression of inflammatory responses”, Du et al. demonstrate that the lncRNA, Mirt2, is induced by TLR4 activation with LPS. Moreover, the authors suggest that mirt2 is a regulator of inflammation through disruption of TRAF6-mediated signaling. The authors suggest that this is due to direct RNA-protein interactions between Mirt2 and TRAF6, inhibiting K63-Ub and thus downstream inflammatory gene activation. Moreover, the authors go on to demonstrate that in vivo overexpression of Mirt2 protects mice from the unabated inflammatory response induced by LPS. Given the lack of knowledge on the regulation and function of Mirt2, this article is of great interest to the field. However, while the data is compelling, due to omission of the experimental details and data quality, the manuscript cannot be accurately reviewed as presented.

Major points:

1) In figure 1, where the p-values derived from the microarray data adjusted for multiple testing? If not, does the significance of differentially expressed genes change? Additionally, it would be beneficial to avoid the use of greenred heatmaps and plots when representing these data. In the volcano plot, how many genes are being represented? Of those, how many of these genes were found to be significantly upregulated/downregulated? Perhaps the authors could consider color coding lncRNAs to distinguish them from coding and other non-coding transcripts as it makes it unclear to visualize mirt2 as the most highly abundant linc.

Re: We thank the reviewer for the suggestions. Multiple testing had not been used in our previous study. We improved the statistical methods and calculated adjusted p-value (q-value) using multiple testing. All of the transcripts in the cluster heatmap and volcano represented long non-coding RNAs. In the volcano plot in our previous figure, 64221 lncRNAs were represented, of those, 2335 were significantly upregulated and 1996

downregulated when filtered with a threshold of fold change ≥ 2 and $p < 0.05$. The differentially expressed genes changed slightly after multiple testing ($q < 0.05$) was employed, that is, 2070 were upregulated (red plots) and 1750 downregulated (blue plots). However, when these differentially expressed lncRNAs were explored by using more stringent criteria ($p < 0.01$, fold change ≥ 20) and filtered according to transcript abundance, as demonstrated in cluster heatmap, the 145 differentially expressed genes (98 upregulated and 47 downregulated) were almost unchanged when using multiple test. In fact, Mirt2 is not the most obviously up-regulated lncRNA even in the cluster heatmap filtered with more stringent criteria. We focused on it because contrast to LPS, IL4 stimulation resulted in a rapid decrease in Mirt2 level. It is this opposite trend that interested us. Moreover, we are also studying on several other lncRNAs and already have preliminary results. The quality of Figure 1a and b has been improved as you suggested and the statistical method and the number of differentially expressed genes were illustrated in the method section and result section respectively.

2) Is the induction of Mirt2 by LPS dose-dependent? The concentrations used for LPS stimulation are quite high. Would the authors expect to see this induction at a lower dose? Moreover, the authors should indicate the concentration of TLR agonists used to assess the induction of Mirt2. According to the Materials and Methods, the authors utilized a student's t-test to evaluate statistical significance, which is not an appropriate statistical test when comparing more than 2 groups.

Re: Our preliminary experimental results showed that the induction of Mirt2 by LPS was dose-dependent, that is way we choose a relatively high concentration. With this concentration of LPS, Mirt2 could be induced remarkably and cells reminded viable. The results were added in Figure 1c. The concentration of TLR agonists were indicated in the figure legend: Pam₃CSK₄ (TLR2/1, 300 ng/mL), Pam₂CSK₄ (TLR2/6, 100 ng/mL), poly(I:C)

(TLR3, 25 $\mu\text{g/mL}$), R848 (TLR7/8, 10 $\mu\text{mol/L}$). The statistical methods have been improved and one-way ANOVA analysis was employed when comparing more than 2 groups. We explained it in revised the method section.

3) In figure 2, the authors should clarify the normalization method for their qPCR data. What reference genes were used for the comparison of relative expression of Mirt2 and other host inflammatory genes? Furthermore, the immunofluorescence images (Fig.2C and E) are too small and low in resolution. Perhaps the authors should zoom in on the cells and provide a quantification of the number of positive cells in the field. Additionally, the authors should clarify the number of replicates used on their FISH experiments.

Re: We appreciate the reviewer's suggestion. The normalization method for the qPCR data was clarified in the revised method section. Total RNA was extracted from cells or tissues with the use of TRIzol reagent (D9108A, Takara Bio). RNA was reverse-transcribed using the RNA PCR Kit (RR036A, Takara Bio). Quantitative polymerase chain reaction (PCR) amplification was performed with an ABI PRISM 7900 Sequence Detector system (Applied Biosystem, Foster City, CA), according to the manufacturer's instructions. Relative gene expression (Mirt2 and other host inflammatory genes, normalized to endogenous control gene β -actin) was calculated using the comparative Ct method formula $2^{-\Delta\Delta C_t}$. The real-time PCR primer sequences are shown in Supplemental table 1. The quality of immunofluorescence images (revised Figure 1e and Figure 2c) was improved as you suggested, and the number of positive cells in the field was quantified (as demonstrated in the lower right of revised Figure 1e and Figure 2c). The FISH experiments in macrophages were repeated three times. For animal experiments, Mirt2 positive cells were counted in four sections per mouse ($n = 6$). We have indicated it in the revised method section or figure legend.

4) Please indicate the concentration of chemical inhibitors used in this study.

Re: Thanks for the reviewer's suggest. We have indicated it in the figure legend.

U0120 (Erk inhibitor): 50 μ M

SP600125 (Jnk inhibitor): 50 μ M

SB203580 (p38 inhibitor): 50 μ M

Bay11-7082 (NF- κ B inhibitor): 10 μ M

PF-04965842 (Jak1 inhibitor): 50 nM

5) In Fig. 3, the authors demonstrate that p38 is crucial for the induction of Mirt2 by showing that chemical inhibition leads to decreased Mirt2 expression. However, has also been shown that p38 is important for the induction of Ifnb in macrophages. Thus, the authors should explore whether this effect is due to decreased Ifnb expression after inhibitor treatment. On the other hand, the authors should show whether or not other TLR stimulations led to robust induction of Ifnb. One would expect that a robust activation of TLR3 would have led to the secretion of type I IFNs and thus activated the phosphorylation of STAT1 and thereby Mirt2 induction.

Re: We appreciate the reviewer's suggestion and have performed additional experiments. Previous studies have shown that p38 is involved in the induction of IFN β in macrophages^{1,2}. However, supplement with exogenous IFN β could not rescue the inhibitory effects of p38 inhibitor on Mirt2 expression (Figure 3h). This means the decreased Mirt2 expression is not due to decreased IFN β after p38 inhibitor treatment. In our previous studies, IFN β stimulation did not affect the basal level of Mirt2 but further increased its expression in LPS-activated macrophages, and these effects could be completely abolished by Stat1 silencing (Figure 3g). Moreover, neutralizing antibody for IFN α/β partly inhibited LPS-induced upregulation of Mirt2 (Figure 3i). The results indicated that LPS-p38-Stat1 was indispensable for the expression of Mirt2, while LPS-IFN α/β -Stat1 enforced these effects. For this reason, replenishment with exogenous IFN β could not rescue the inhibitory effects of p38 inhibitor on Mirt2 expression, since LPS-p38-stat1 pathway reminded repressed.

To further explore the mechanism, we detected the induction of IFN β and the activation of Stat1 upon TLRs stimulation. Our studies showed that engagement of TLR2 (Pam₃CSK₄, Pam₂CSK₄), TLR4 (LPS) and TLR7/8 (R848) led to rapid phosphorylation (30 min) of Stat1 at serine 727 (S-727) in murine macrophages, whereas TLR4 (LPS), TLR3 (poly(I:C)) and TLR7/8 (R848) induced Stat1 phosphorylation at tyrosine 701 (T-701), although this response was delayed (4 hours) compared with S-727 phosphorylation (Figure 3k). Previous studies have shown that TLR-induced Stat1 serine phosphorylation was dependent on p38, however, tyrosine phosphorylation of Stat1 is indirectly mediated by the production of endogenous type I IFNs, particularly IFN β ³. Consistently, we found that TLR4, TLR3 or TLR7/8 stimulations led to robust induction of IFN β (Figure 3j). Our previous results showed that only TLR2, TLR4 and TLR7/8 stimulations led to induction of Mirt2 (Supplemental figure 1b), this indicated that p38-Stat1 (S-727) was indispensable

for the induction, while IFN α/β -Stat1 (T-701) could enforce these effects. The possible mechanisms were depicted in Supplemental figure 6.

Supplemental figure 1

Supplemental figure 6

6) While the authors claim that there are no changes in the expression of TLR4 and/or other related adaptor molecules following silencing or overexpression of Mirt2, the data as presented (SFig.3 F-H) is hard to interpret due to contrast/exposure issues.

Re: The quality of WB images in Supplemental figure 3 has been improved.

7) The authors should provide better quality WB images for figure 3. The images presented are mostly underexposed. Additionally, the p65 nuclear translocation figure is too small to observe changes in the relative amount of translocation. Perhaps the authors could show a representative zoomed in image and quantify the number of events where there is decreased translocation. As mentioned before, the authors should provide details as to the number of replicates these images represent.

Re: The quality of WB images in Figure 4 has been improved. The p65 nuclear translocation figure has been replaced and quantified, as demonstrated in Figure 4h and i. These images represent three replicates of experiments, and we indicated it in the figure legend.

8) In figure 5, the authors should avoid the use of increased contrast or underexposure of blots in order to accurately review the data. Additionally, do the authors see colocalization between Mirt2 and TRAF6 in non-stimulated macrophages? Is the

endogenous interaction dependent of stimulation? Please indicate whether the images in Fig. 5D correspond to a single slice of Z-stacks or MFI representations. Also indicate the number of field/replicates used to determine these interactions. Given the high concentration of LPS utilized in this study, the authors should include a housekeeping gene to demonstrate LPS-treated cells are viable.

Re: The quality of WB images in Figure 5 has been improved. We also observed the colocalization between Mirt2 and TRAF6 in non-stimulated macrophages (Figure 5d), which indicated that the endogenous interaction was not dependent of stimulation. However, since the basal level of Mirt2 was relatively low and TRAF6 was activated upon stimulation, the roles of Mirt2 in resting cells could be limited, or need further study. The images in Figure 5d correspond to a single slice of Z-stacks and the experiments were repeated three times. We indicated in the figure legend. The cell viability was confirmed by MTT assay.

9) Does Mirt2 expression affect the signal transduction of other receptors upstream of TRAF6?

Re: We appreciate the reviewer's suggestion and have performed additional experiments. Besides Toll-like receptor family, TRAF6 is also a crucial docking molecule that mediates signaling events initiated by interleukin-1 (IL-1) receptor family and tumor necrosis factor (TNF) receptor family (such as receptor activator of NF- κ B, RANK) in macrophages^{4,5}. As demonstrated in Supplemental figure 3i, the phosphorylation of p65 activated by IL1 β or receptor activator of NF- κ B ligand (RANKL, also known as OPGL or ODF) was significantly inhibited by Mirt2. However, Mirt2 had no effects on TNF- α induced activation of p65, which mediated by TRAF2.

10) In line 576, the authors acknowledge that there is no human homologue of Mirt2. However, they indicate that murine-derived Mirt2 might have anti-inflammatory in human cells. The authors should include this data as Mirt2 might have therapeutic potential in the regulation of inflammation.

Re: The anti-inflammatory effects of Mirt2 in human monocyte-derived macrophages and hepatocytes were confirmed. The results were demonstrated in Supplemental figure 5f and g.

11) In SFig.3, it seems that the phosphorylation of TAK1 is diminished after overexpression of Mirt2. Have the authors explored the interaction between TAK1 and Mirt2? One would expect that the Ub of TRAF6 would be impacted.

Re: We appreciate the reviewer's suggestion and performed a pull-down assay with biotinylated Mirt2, followed by immunoblot with anti-TAK1 antibody. As demonstrated in Supplemental figure 3h, we failed to detect the interaction between TAK1 and Mirt2, and we considered that the diminished phosphorylation of TAK1 was due to the inhibition of

TRAF6 ubiquitination by Mirt2.

h

12) KD efficiencies for all the siRNAs should be shown by western blot.

Re: We appreciate the reviewer's suggestion and KD efficiencies for siRNAs have been shown by western blot, as demonstrated in Figure 3d, Supplemental figure 1f and i.

Minor points:

1) The manuscript needs to be revised for English grammar and clarity. Please change mirt2 to Mirt2.

Re: Thanks for your comments. We have improved the manuscript as you suggested.

2) If possible, please use bigger fonts and enlarge the figures as it is hard to evaluate the data and read the legends. This is especially important for the evaluation of microscopy/histology data.

Re: Thanks for your comments. The quality of figures has been improved in the revised manuscript.

3) Please include in the Materials and Methods information regarding the antibodies used for imaging and the secondary antibodies used. Please indicate the amino acid residues evaluated for phosphorylation.

Re: Thanks for your comments. The information of antibodies has been included in the

method section, and the amino acid residues evaluated for phosphorylation were indicated in related figures.

Reviewer #2 (TRAF6 signalling) (Remarks to the Author):

The manuscript from Du et al have identified mirt2, a long non-coding RNA as a negative regulator of inflammatory responses, via its inhibitory effects on TRAF6. By searching for LPS-induced long non-coding RNA the authors have identified mirt2 as a target, which is expressed mainly the cytoplasm in activated macrophages from septic mice. They provide also experimental evidence for that p38-Stat1 and IFN-Stat1 signaling pathways regulates expression of mirt2. Adenovirus-induced overexpression of mirt2 in mice, suppress LPS-induced inflammatory responses in macrophages. Interestingly, mirt2 associates to the Ring-domain and Zn-fingers in TRAF6 and can thereby prevent the catalytic activity of TRAF6. In detailed in vivo experiments performed in mice, the authors provide detailed and convincing data for a physiological role of mirt2 to regulate inflammatory responses and polarization of macrophages. Moreover and interestingly, mirt2 is found to be specifically expressed in M1 macrophages in pathological conditions such as atherosclerosis, cardiac infarction, diabetic nephropathy and obesity.

The experimental design is very good and the presented data are convincing and provides a molecular understanding for the regulation of mirt2, its role in sepsis and pathological conditions and its inhibitory effects on TRAF6. Why the kinetics for regulation of mirt2 is relatively slow is interesting but could represent a physiological response to counteract acute inflammatory responses.

Minor comments:

Figure 4h, nuclear p65 should be quantified.

Re: Thanks for your comments. The p65 nuclear translocation figure has been replaced and quantified, as demonstrated in Figure 4h and i.

Figure 7d, the quality of this figure could be improved.

Re: The Figure 7d was replaced by our repeated experimental results.

Figure 9. The schematic diagram is good for the reader to more easily access the presented data. The big red arrow for mirt2 in it's center could maybe be reduced in size and the long bended arrow in the left for IFN alpha/beta could be straight lines as well. PPARgamma should stand together.

Re: Thanks for your comments. We have improved it in the revised manuscript.

Main text: There are some minor mistakes in the main text on line 102,435, 541 and 542. Please check.

Re: Thanks for your comments. We have improved it in the revised manuscript.

Statistical analyses: The authors could consider to use other statistical analyses such as ANOVA, to strengthen their report.

Re: Thanks for your comments. The statistical methods have been improved and one-way ANOVA analysis was employed when comparing more than 2 groups. We explained it in the method section.

Reviewer #3 (Sepsis and TLR4 signalling) (Remarks to the Author):

1. The definition of sepsis is infection with organ dysfunction due to a dysregulated immune response. Treating animals with LPS is not sepsis but instead is endotoxemia. All reference to sepsis should be removed from the paper and the figures when LPS is used.

Re: Thanks for your comments. We have improved it in our revised manuscript.

2. The results suggest that TLR-induced type 1 IFN drives Mirt2 expression. Is this MyD88 or TRIF-dependent? This is important because the induction by some of the agonists would be expected to drive signaling through MyD88 and not TRIF; however, TRIF signaling is more prominent in type 1 interferon expression.

Re: Thanks for your comments. The induction of Mirt2 by LPS was partly inhibited by knockdown of Myd88 or TRIF, and knockdown both completely abrogated the effects of LPS (Supplemental figure 1e). As our previous study demonstrated, LPS-p38-Stat1 was indispensable for the expression of mirt2, and LPS-IFN α / β -Stat1 enforced these effects. TRIF signaling is more prominent in type 1 IFNs expression, while p38 activation is through Myd88 or TRIF pathway independently^{6,7}. Knockdown of TRIF abrogated the induction of type 1 IFNs, but with no effects on p38 phosphorylation. While knockdown of Myd88 could not completely inhibit the activation of p38. We consider that is way only knockdown both could completely abrogate the induction of Mirt2 by LPS. Besides, we have performed additional experiments to explore the mechanisms on Mirt2 expression, which were depicted in Supplemental figure 6.

Supplemental figure 1

Supplemental figure 6

3. In figure 2E, there is an impressive elevation of Mirt2 in the serum. What is the meaning of this? Can systemic administration of Mirt2 lead to suppression of TLR4 signaling? Is Mirt2 an intercellular signaling molecule? Understanding this would elevate the significance of the observation and add to the paper.

Re: Thanks for your comments. We have explained it in discussion section. Noteworthy, we identified that the level of Mirt2 was dramatically increased in the plasma of endotoxemia mice (Figure 2b). In light of our studies, we propose that this release could be attributable to the increased levels of Mirt2 in the diseased organs. However, adenovirus mediated gene transfer could not further lead to a rise in plasma Mirt2 level (Figure 6a). This may suggest that the release of Mirt2 is under a stringent control mechanism and simply increasing Mirt2 expression via an adenovirus approach is not sufficient to induce the additional release of this lncRNA from the cells. Whether plasma Mirt2 is simply a by-product of increased intracellular levels or is functionally active in disease pathology still remains unclear, and this needs our further study. In fact, we have confirmed that Mirt2 existed in circulating exosomes and increased in endotoxemia mice. Exosomes are now viewed as specifically secreted vesicles that enable intercellular communication and have become the focus of exponentially growing interest. It is reasonable to assume that Mirt2 may promote the cross-talk between various organs via this new way about cell signaling. However, due to limited conditions and lack of experience, we did not carry on further studies at this stage. Since adenovirus mediated gene transfer could not further lead to a rise in plasma Mirt2 level, and systemic administration of Mirt2 seemed infeasible (unlike microRNA, another kind of non-coding RNA, lncRNA synthesized by in vitro transcription is unstable and unfit for animal experiments), the intercellular roles of Mirt2 have not been fully studied in the present study. No doubt it is a new exploration that needs our further work.

4. In figure 4, the authors should provide evidence that Mirt is either over or under expressed by the adenoviral vectors.

Re: We appreciate the reviewer's suggestion and the results have been shown in Supplemental figure 2j.

5. In figure 4H, there does not appear to be any difference induced by Mirt2 transduction in P65 translocation.

Re: The p65 nuclear translocation figure has been replaced and quantified, as demonstrated in Figure 4h and i.

6. In figure 6A, the authors need to give the time point for the measurement of Mirt2. It would be important to be sure to have measured Mirt2 expression at 72 hours, the time point that LPS was given. Why wait 72 hours?

Re: Thanks for your comments. We have indicated it in the figure legend. In Figure 6a, we measured Mirt2 level at 72 hours after adenovirus administration, the time point that LPS was given. Time-course experiments revealed that transgene expression began at 24 hours, gradually increased and peaked at 72 hours, then declined after that. For the optimal expression efficiency, LPS was given at 72 hours after adenovirus administration.

7. What is the circulating levels of Mirt2 after the AdMirt2 transduction?

Re: As demonstrated in Figure 6a, the serum Mirt2 level after AdMirt2 transduction was

not increased significantly compared to control group. This may suggest that the release of Mirt2 is under a stringent control mechanism and simply increasing mirt2 expression via an adenovirus approach is not sufficient to induce the additional release of this lncRNA from the cells.

8. It would be important to remove cells such as macrophages from the animals receiving the AdMirt2 and test their responsiveness in vitro to assure that Mirt2 transduction blocked LPS responsiveness in one of the key target cells.

Re: We appreciate the reviewer's suggestion and have performed additional experiments. Adenovirus (AdMirt2 or Ad-EV) were delivered into mice by tail vein injection. After 3 days, primary peritoneal macrophages were prepared, cultured in vitro and challenged with LPS for 4 and 20 hours. Compared to control group, the expression of inflammatory factors in macrophages from AdMirt2 treated mice was significantly inhibited (Supplemental figure 2i-k).

9. It would be helpful for the authors to provide the concentrations of the LPS used and the time points studied in the text.

Re: Thanks for your comments. We have improved it in the revised manuscript.

10. In figure 8, the time point of Mirt2 measurement relevant to the age of the mice or the time in the course of the disease process needs to be provided.

Re: We have illustrated these issues in figure legend.

a, Male C57BL/6J and ApoE^{-/-} mice were fed a Western diet from 6 to 18 weeks old, then the mice were sacrificed and Mirt2 in aorta was detected.

b, Male C57BL/6J mice aged 8 weeks were suffered left-anterior descending coronary artery (LAD) ligation, and sham group as control. 24 hours after operation, mice were sacrificed and Mirt2 in myocardium was detected.

c, Male C57BL/6J mice aged 12 weeks were injected intraperitoneally with a single dose of CCl₄ (1 ml/kg) or an equal volume of vehicle. After 72 hours, mice were sacrificed and Mirt2 in liver was detected.

d, Male C57BL/6J mice aged 6 weeks were fed with standard chow diet (SCD) or high fat diet (HFD) containing 60% calories as fat for 12 weeks. Then mice were sacrificed and Mirt2 in liver was detected.

e, Male C57BL/6J mice aged 10 weeks were fed with standard chow diet (SCD) or methionine-choline deficient diet (MCD) diet for 6 weeks. Then mice were sacrificed and Mirt2 in liver was detected.

f, Male C57BL/6J and ob/ob^{-/-} mice were sacrificed at 24 weeks of age and Mirt2 in adipose tissue was detected.

g and h, Male C57BL/6J mice aged 10 weeks were injected intraperitoneally with 50 mg/kg/day STZ for five consecutive days. Age-matched control mice received an equal volume of vehicle. One week after STZ injection, fasting blood glucose was checked, and mice with blood glucose > 300 mg/dL were used for experiments. The mice were sacrificed at 24 weeks of age and Mirt2 in kidney and myocardium was detected.

References:

1. McGuire, V.A., *et al.* Beta Interferon Production Is Regulated by p38 Mitogen-Activated Protein Kinase in Macrophages via both MSK1/2- and Tristetraprolin-Dependent Pathways. *Molecular and cellular biology* **37**(2017).
2. Navarro, L. & David, M. p38-dependent activation of interferon regulatory factor 3 by lipopolysaccharide. *The Journal of biological chemistry* **274**, 35535-35538 (1999).
3. Rhee, S.H., Jones, B.W., Toshchakov, V., Vogel, S.N. & Fenton, M.J. Toll-like receptors 2 and 4 activate STAT1 serine phosphorylation by distinct mechanisms in macrophages. *The Journal of biological chemistry* **278**, 22506-22512 (2003).
4. Lamothe, B., *et al.* The RING domain and first zinc finger of TRAF6 coordinate signaling by interleukin-1, lipopolysaccharide, and RANKL. *The Journal of biological chemistry* **283**, 24871-24880 (2008).
5. Strickson, S., *et al.* Roles of the TRAF6 and Pellino E3 ligases in MyD88 and RANKL signaling. *Proceedings of the National Academy of Sciences of the United States of America* **114**, E3481-E3489 (2017).
6. Jeyaseelan, S., *et al.* Toll/IL-1 receptor domain-containing adaptor inducing IFN-beta (TRIF)-mediated signaling contributes to innate immune responses in the lung during Escherichia coli pneumonia. *J Immunol* **178**, 3153-3160 (2007).
7. Kirkwood, K.L. & Rossa, C., Jr. The potential of p38 MAPK inhibitors to modulate periodontal infections. *Current drug metabolism* **10**, 55-67 (2009).

REVIEWERS' COMMENTS:

Reviewer #3 (Remarks to the Author):

The authors have addressed all of my concerns. The work is an important advance in the field and should guide other labs to perform work in related areas of inflammation research. The highly specific role of dirt 2 is impressive and also could be exploited for therapeutic purposes. The authors work both in vitro and in vivo and across a number of relevant cells types. Therefore the reproducibility and relevance appears to be high.